# Two-Layer Linear Auto-Regressive Models Estimate Latent States

Yahya Sattar [1]   Sunmook Choi [1]   Leo Maynard-Zhang [2]   Yassir Jedra [3]   Maryam Fazel [2 4]   Sarah Dean [1]

## Abstract

Auto-regressive models have emerged as powerful tools for sequential data, from language to video. Understanding how and why these models learn latent representations remains an open theoretical question. In this work, we demonstrate that when trained by empirical risk minimization on data from partially observed linear dynamical systems, two-layer linear auto-regressive models naturally learn to approximate Kalman filtering. In particular, we show that the learned hidden representation coincides, up to a similarity transformation, with the state estimates produced by the optimal (Kalman) filter, even though the model has no explicit knowledge of the underlying dynamics or state. The result follows from three main insights. First, we establish that the Kalman filter is well approximated by an auto-regressive model with bounded truncation error. Second, we show that despite non-convexity, the two-layer optimization landscape is benign, i.e., all stationary points are either strict saddles or global minima. Finally, as our main contributions, we provide finite-sample guarantees on prediction error, parameter estimation error, and latent state recovery. Numerical simulations support the theoretical results and demonstrate that the latent representations of auto-regressive models recover state estimates.

## 1. Introduction

Fueled by sequential data, auto-regressive models have emerged as powerful general purpose tools. Examples range from large language models (LLMs) trained on internet scale text data to world models trained with robot video streams. The prevailing approach for leveraging this data is to train models which predict the next element of a sequence given past elements. Whether these capable models also learn the deeper mechanisms underlying the data is an open research question. Empirical findings on large foundation models are mixed; there is evidence that LLMs represent the board state when given a sequence of chess moves (Li et al., 2023; Toshniwal et al., 2022), but also that they confuse states which have equivalent sets of legal moves (Vafa et al., 2025). The goal of learning good representations predates the current moment. It has been a key challenge and motivation for deep learning since the beginning, with major successes like word2vec for modeling language (Church, 2017) and matrix factorization for movie recommendation (Funk, 2006; Koren et al., 2009).

The connection between observables (inputs and outputs) and latent states has historically been the domain of dynamical systems and control theory (Willems, 1989), especially for time-indexed data. In particular, the field of system identification has long investigated what system properties can be distinguished from input-output data alone. Over the last decade, this classical theory has been revisited and modernized from a statistical learning perspective (Dean et al., 2019; Simchowitz et al., 2018). Of particular relevance is a line of work on finite-sample theory for learning models of linear dynamical systems when the latent state is only partially observed (Oymak & Ozay, 2019; Bakshi et al., 2023; Sarkar et al., 2021). These works largely rely on shallow (linear) models and, to go from observables to latent states, they rely on classical approaches like the Ho-Kalman factorization (Ho & Kálmán, 1966) or nuclear norm regularization (Recht et al., 2010; Sun et al., 2022). These techniques directly search for the latent state or introduce special-purpose regularization, which stands in contrast to the end-to-end training paradigm dominant in deep learning.

In this paper, we unite the perspective of dynamical systems theory with the standard end-to-end deep learning approach. We draw on a rich line of related work on finite-sample learning for linear dynamical systems and recent developments in the theory of non-convex optimization for matrix factorization, further discussed in Section 2. Our theory shows that two-layer linear auto-regressive models naturally learn to approximate Kalman filtering when trained by empirical risk minimization on data from partially observed linear dynamical systems. Our key contributions are as follows:

[1]Cornell University [2]University of Washington Seattle [3]Imperial College London [4]Amazon Inc. Correspondence to: Yahya Sattar <ysattar@cornell.edu>, Sarah Dean <sdean@cornell.edu>.

*Proceedings of the 43rd International Conference on Machine Learning*, Seoul, South Korea. PMLR 306, 2026. Copyright 2026 by the author(s).

- We formulate a novel two-layer auto-regressive model for learning to estimate latent states of a partially observed linear dynamical system (Section 3).

- We show that despite the non-convexity of the learning objective, the optimization landscape is benign, i.e., all stationary points are either strict saddles or global minima (Section 4.2).

- We provide finite-sample guarantees (sample complexity, and statistical error rates) on prediction error and parameter estimation error (Section 4.3) and then show that these imply latent state recovery (Section 4.4).

We conclude with numerical simulations which demonstrate that auto-regressive models automatically represent latent state estimates (Section 5) and a discussion of broader implications and directions for future work (Section 6).

## 2. Related Work

We build on a line of work concerned with finite sample identification of linear dynamical systems. The prevalent strategy (when the state is not directly observed) is to first learn a linear auto-regressive model, and then to use a classical factorization technique (Ho & Kálmán, 1966) to extract the state space parameters. Oymak & Ozay (2021) present a modern perturbation analysis of this approach. In contrast, we do not separate learning a model from extracting information about the latent state; we show that the latent state estimate naturally arises in the activations of a two-layer linear model.

Much of the prior work is focused on estimating the linear parameters through regression: either predicting the next output from previous inputs (Oymak & Ozay, 2019; Sun et al., 2022; Sarkar et al., 2021) or, in true auto-regressive fashion, predicting the next output from previous inputs and outputs. The latter allows for consistent estimation of marginally stable systems (i.e., those without decaying memory) or uncontrolled systems (i.e., those without observed inputs), but must handle more complex dependencies in the covariates. For this setting, there are a range of estimation techniques based on pre-filtering (Simchowitz et al., 2019; Bakshi et al., 2023), spectral filtering (Dogariu et al., 2025), or simple linear regression (Tsiamis & Pappas, 2019; Lale et al., 2020; Lee & Lamperski, 2020). The last category is most closely related to the present paper, since we also consider an auto-regressive least-squares learning objective. It is worth particularly highlighting Tsiamis & Pappas (2019) who presented the first non-asymptotic analysis of the simple regression approach and, similar to us, focus on Kalman filtering, rather than parameter estimation or downstream control tasks. However, while all of the aforementioned works follow the prevalent estimate-then-decompose strategy, we propose and analyze a non-convex

learning procedure. For this, we leverage statistical tools developed by (Ziemann et al., 2022; Ziemann & Tu, 2022) for characterizing system identification errors at the global optima of learning objectives, regardless of their convexity.

There are a handful of results on non-convex approaches for learning linear dynamical systems, most of which do not consider auto-regressive architectures. Hardt et al. (2018) analyzed gradient descent on a *recurrent* linear model whose parameters are exactly state space parameters. Tadipatri et al. (2025) propose two nonconvex reformulations for low-order system identification in terms of both state space parameters and a factorized Hankel matrix. The latter bears similarity to our learning objective, though while we show that our optimization landscape is benign, Tadipatri et al. (2025) focus on non-convex optimization algorithms. Umenberger et al. (2022) analyze policy gradient methods for directly learning the static gains of the Kalman filter without model identification.

Auto-regressive policies have appeared in the literature in the context of optimal linear control. For quadratic costs, the optimal policy is the composition of a Kalman filter with a linear state feedback controller. Both classical work (Skelton & Shi, 1994) and more recent results (Al Makdah et al., 2022; Guo et al., 2023) show how to represent Kalman filters auto-regressively for control, but do not consider statistical aspects. In policy learning, where explicit model identification is not required, auto-regressive policies exhibit a more benign optimization landscape than standard parametrization (Fallah et al., 2025; Zhao et al., 2023; Xie & Ni, 2024). Like these works, we formulate an auto-regressive representation of the Kalman filter, but unlike them, we use a two-layer model and focus on state estimation rather than control.

Finally, we share a similar spirit to some recent work which makes connections between linear systems and modern machine learning practice, but which is otherwise quite different in terms of goals and techniques. Inspired by "world models" and representation learning techniques in reinforcement learning, Tian et al. (2023a;b) use additional supervision from the cost signal to learn latent states in the context of linear quadratic optimal control. In another vein, Goel & Bartlett (2024) show that an auto-regressive Transformer can approximate a Kalman filter for any given linear dynamical system, while Du et al. (2023) investigate the ability of Transformers to generalize across multiple linear systems.

## 3. Setting: Linear Dynamics and Filter

Consider a partially observed linear dynamical system with the following state-space representation: for all $t \geq 0$,

$$\begin{aligned} x_{t+1} &= Ax_t + Bu_t + w_t, \\ y_t &= Cx_t + v_t, \end{aligned} \tag{3.1}$$

where at time $t$, $x_t \in \mathbb{R}^n$ represents the latent state, $u_t \in \mathbb{R}^p$ is the observed control input, $y_t \in \mathbb{R}^m$ is the measured output, $w_t \in \mathbb{R}^n$ is the unobserved process noise, and $v_t \in \mathbb{R}^m$ is the unobserved measurement noise.

**Assumption 1.** *During the training period, the noise processes $\{w_t\}_{t\geq0}$, $\{v_t\}_{t\geq0}$ and the excitations $\{u_t\}_{t\geq0}$ are sequences of independent, zero-mean, Gaussian random vectors with covariance $\Sigma_w\succ0$, $\Sigma_v\succ0$, and $\Sigma_u\succ0$, respectively.*

Note that we do not assume knowledge of the matrices $A, B, C$ or the noise covariances $\Sigma_w, \Sigma_v$. Given only a single trajectory of input-output samples $\{(u_t, y_t)\}_{t=0}^{T+H}$ from the system (3.1) satisfying Assumption 1, our goal is to directly learn the optimal filter, without explicitly learning the system parameters $A, B, C$, and $\Sigma_w, \Sigma_v$.

Before moving onto the filtering problem, we conclude this section by reviewing a classic result of linear system theory. Consider any invertible matrix $S$, which we term a similarity transform. Then there is an equivalence class of state space representations which cannot be distinguished from input/output data alone.

**Remark 1.** *A system with parameters $A, B, C$, and $\Sigma_w, \Sigma_v$ will produce identical input-output statistics to a system with parameters $SAS^{-1}, SB, CS^{-1}$, and $S\Sigma_wS^\top, \Sigma_v$ for any similarity transform $S$. If the first system has latent state $x$, then the alternative representation will have latent state $\tilde{x} = Sx$, and both are equally valid state space representations of the same input-output behavior.*

**Notation:** The $\ell_2$-norm of a vector $x$ is denoted by $\|x\|_{\ell_2}$. The spectral radius, the spectral norm, and the Frobenius norm of a matrix $X$ are denoted by $\rho(X), \|X\|$, and $\|X\|_F$, respectively. The largest and smallest eigenvalue of a square matrix $X$ are denoted by $\lambda_{\max}(X)$ and $\lambda_{\min}(X)$. $\sigma_i(X)$ denotes the $i$-th singular value of a matrix $X \in \mathbb{R}^{m \times n}$ with $\sigma_1(X)$ being the largest and $\sigma_n(X)$ being the smallest (when $n\leq m$). Given a sequence of vectors $\{x_t\}_{t=1}^T$, we denote by $x_{t:t+k}$, the column-wise concatenation of $x_t, \ldots, x_{t+k}$. For a centered random vector $z$, we use $\Sigma[z] = \mathbb{E}[zz^\top]$ to denote its covariance matrix. $\otimes$ denotes the Kronecker product Lastly, $\tilde{\mathcal{O}}$, $\lesssim$ and $\gtrsim$ hide constants and logarithmic terms involving the problem variables.

### 3.1. Kalman Filter

For a given dynamics model, the Kalman filter is the best linear filter for predicting the latent states, and when the noise processes are Gaussian, it is the optimal filter. Given the system parameters $A, B, C, \Sigma_w, \Sigma_v$, the steady-state Kalman filter in its predictor form is given by

$$\begin{aligned} \hat{x}_{t+1} &= \bar{A}\hat{x}_t + Bu_t + Fy_t, \\ y_t &= C\hat{x}_t + e_t, \end{aligned} \quad (3.2)$$

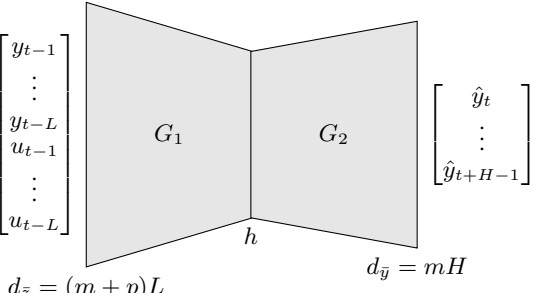

$d_{\bar{z}} = (m+p)L \qquad\qquad d_{\bar{y}} = mH$

*Figure 1.* Two-layer auto-regressive model architecture

where $\hat{x}_t$ is the predicted state estimate at time $t$, and $\bar{A} := A - FC$ is the closed-loop estimator matrix. The Kalman gain matrix $F$ depends on $\Sigma$, the solution to a discrete-time algebraic Riccati equation (Tian et al., 2023b), and also the steady-state error covariance of the Kalman filter. In general, depending on the initial state covariance $x_0 \sim \mathcal{N}(0, \Sigma_0)$, the Kalman gain matrix is time varying. However, under standard conditions, it converges exponentially fast to the static gain $F$ (Komaroff, 2002). We therefore consider only the steady-state Kalman filter. This is common in the literature (Tsiamis & Pappas, 2019; Lale et al., 2020). It corresponds to assuming that the initial state covariance $\Sigma_0 = \Sigma$ is the solution to the Riccati equation. Under Assumption 1, at steady state the so-called innovation term $e_t$ is distributed as $\{e_t\}_{t=0}^{T+H} \overset{\text{i.i.d.}}{\sim} \mathcal{N}(0, \Sigma_e)$, where $\Sigma_e = C\Sigma C^\top + \Sigma_v$ (Anderson & Moore, 2005).

### 3.2. Auto-regressive Model

With the objective of learning the Kalman filter directly from the data, we train a two-layer linear auto-regressive model on a single trajectory $\{(u_t, y_t)\}_{t=0}^{T+H}$ of (3.1). Specifically, for a selected history length $L>0$, we construct the covariates $\{\bar{z}_t\}_{t=1}^T$ as follows,

$$\bar{z}_t := [y_{t-1}^\top \cdots y_{t-L}^\top u_{t-1}^\top \cdots u_{t-L}^\top]^\top \in \mathbb{R}^{d_{\bar{z}}}, \quad (3.3)$$

which are inputs to our auto-regressive model. Here we set $d_{\bar{z}} := (m+p)L$, and use $\{u_t\}_{t\leq-1} = \{y_t\}_{t\leq-1} = 0$. Similarly, for a fixed future horizon $H>0$, the output prediction of the auto-regressive model is $y_{t:t+H-1} \in \mathbb{R}^{d_{\bar{y}}}$, where we set $d_{\bar{y}} := mH$. These covariates and predictions define the input and output dimensions of the auto-regressive model. We furthermore consider two-layer models with hidden dimension $h$, as illustrated in Figure 1. Formally, the function class $\mathcal{F} := \{f(\bar{z}) = G_2G_1\bar{z} : (G_1, G_2) \in \mathcal{G}(h)\}$ is defined by

$$\begin{aligned} \mathcal{G}(h) := \{ &(G_1, G_2): G_1 \in \mathbb{R}^{h \times d_{\bar{z}}}, \ G_2 \in \mathbb{R}^{d_{\bar{y}} \times h}, \\ &\max\{\|G_1\|_F^2, \|G_2\|_F^2\} \leq c_0 \}. \end{aligned}$$

We train this model on the constructed covariate-output pairs $\{(\bar{z}_t, y_{t:t+H-1})\}_{t=1}^T$ with a squared loss:

$$\mathcal{L}_{\mathcal{R}}(G_1, G_2) := \frac{1}{2T} \sum_{t=1}^{T} \|y_{t:t+H-1} - G_2 G_1 \bar{z}_t\|_{\ell_2}^2 \quad (3.4)$$

The training objective is the following empirical risk minimization (ERM) problem,

$$(\hat{n}, \hat{G}_1, \hat{G}_2) \in \operatorname*{argmin}_{h \leq r, (G_1, G_2) \in \mathcal{G}(h)} \mathcal{L}_{\mathcal{R}}(G_1, G_2) \quad (3.5)$$

We solve the non-convex ERM problem (3.5) by performing: (i) architecture search over the hidden state dimension $h$, and (ii) gradient descent over $G_1$, and $G_2$ for each fixed $h$. We will discuss the optimization landscape in more detail in Section 4.2. We remark on the connections between this objective and deep learning practice. The optimization over $G_1$ and $G_2$ corresponds to training a two layer linear network, where the weights are bounded (due to the norm bound in the definition of $\mathcal{G}(h)$). Solving an optimization problem with bounded parameters is equivalent to solving one with a regularized objective, where the regularization penalty corresponds to some $c_0$ (Hastie et al., 2009). In deep learning practice, it is common to implement the regularization as "weight decay" in the optimization algorithm. Hence, in practice, the inner optimization of (3.5) can be implemented by training a two layer network with linear activations and weight decay. We will discuss the optimization landscape of (3.5) in more detail in Section 4.2.

## 4. Main Results

In this section, we state our key positive result that a two layer network trained auto-regressively learns to perform Kalman filtering. First, we introduce some conditions required for our results to hold. These conditions are very standard in subspace identification literature (Knudsen, 2001).

**Definition 1.** *The matrix pair $(A, C) \in (\mathbb{R}^{n \times n}, \mathbb{R}^{m \times n})$ is observable if $[C^\top \ (CA)^\top \ \cdots \ (CA^{n-1})^\top]^\top$ has full column rank. The matrix pair $(A, B) \in (\mathbb{R}^{n \times n}, \mathbb{R}^{n \times p})$ is controllable if $[B \ AB \ \cdots \ A^{n-1}B]$ has full row rank. Lastly, the pair $(A, B)$ is stabilizable if there exists a matrix $K$ such that $\rho(A - BK) < 1$.*

**Assumption 2.** *(a) The system (3.1) is non-explosive, i.e., $\rho(A) \leq 1$; (b) The pair $(A, \Sigma_w^{1/2})$ is stabilizable, and the pair $(A, C)$ is observable.*

Assumption 2(a) ensures that covariates will not become poorly conditioned due to large outputs, whereas 2(b) guarantees that a steady-state Kalman filter for the system (3.1) exists, and furthermore that the filter is stable $\rho(A - FC) < 1$. Stability of the filter closed loop $\bar{A} = A - FC$ plays a crucial role in our truncation argument. Recall that,

by Gelfand's formula, for all $\rho > \rho(\bar{A})$, the quantity $C_\rho := \sup_{k \in \mathbb{Z}_+} (\|\bar{A}^k\|/\rho^k)$ is finite, and it is known to be $C_\rho \geq 1$. Hence, if $\rho(\bar{A}) < 1$, then for all $\rho \in (\rho(\bar{A}), 1)$, we can bound the powers $\|\bar{A}^k\| \leq C_\rho \rho^k$ for all $k \in \mathbb{Z}_+$.

The following statement summarizes our main results on the auto-regressive learning of Kalman filtering from data in Theorem 4.

**Theorem 1** (Main result – Informal). *Let $(\hat{G}_1, \hat{G}_2)$ be the global minimizer of the ERM problem (3.5). Suppose Assumptions 1, 2 hold. For any new sequence of observed inputs-outputs $\{(u_\tau, y_\tau)\}_{\tau=0}^{t-1}$, let $\hat{x}_t$ be the Kalman filter estimate, and $\bar{z}_t$ be the covariate. Then, there exists a similarity transform $S$, and a scalar $\beta > 0$, such that choosing*

$$L \gtrsim \beta \log(T)/(1 - \rho),$$
$$and, \quad T \gtrsim L\left(d_{\bar{z}} + \log(T/\delta)\right),$$

*with probability at least $1 - \delta$, we have*

$$\left\|\hat{x}_t - S\hat{G}_1\bar{z}_t\right\|_{\ell_2}^2 \lesssim \frac{H}{T}\left(r\left(d_{\bar{y}} + d_{\bar{z}}\right) + \log\left(\frac{T}{\delta}\right)\right)\|\bar{z}_t\|_{\ell_2}^2$$

Theorem 1 shows that a two-layer auto regressive model approximately performs Kalman filtering at the hidden layer. In other words, a linear auto-regressive model trained only on input-output data automatically approximates the estimates of the best linear filter which knows the underlying model. The gap between the Kalman filter estimate $\hat{x}_t$ and $S\hat{G}_1\bar{z}_t$ decays with the size of training data as $\tilde{\mathcal{O}}(1/\sqrt{T})$. In the remainder of this section, we will state some intermediate results, leading to our main result on latent state recovery.

### 4.1. Filter Approximation Results

In this section, we show that under Assumption 2, the true Kalman filter can be approximated by a linear function of $L > 0$ past inputs and outputs. The approximation error depends on the history length $L$ and the spectral radius of $\bar{A}$, i.e. the filter stability. Specifically, expanding the Kalman Filtering predictor form (3.2), we can express the estimated state in terms of the covariate $\bar{z}_t$ as follows,

$$\hat{x}_t = \mathcal{C}\bar{z}_t + \bar{A}^L \hat{x}_{t-L}, \quad (4.1)$$

The matrix $\mathcal{C} \in \mathbb{R}^{n \times (m+p)L}$ is the extended controllability matrix, defined as

$$\begin{aligned}
\mathcal{C}_y &:= \begin{bmatrix} F & \bar{A}F & \cdots & \bar{A}^{L-1}F \end{bmatrix}, \\
\mathcal{C}_u &:= \begin{bmatrix} B & \bar{A}B & \cdots & \bar{A}^{L-1}B \end{bmatrix}, \\
\mathcal{C} &:= \begin{bmatrix} \mathcal{C}_y & \mathcal{C}_u \end{bmatrix}.
\end{aligned}$$

This matrix maps the effects of inputs and outputs on the estimated state.

**Proposition 1.** *Fix a time index $t \geq L$, and a failure probability $\delta \in (0,1)$. Then, under Assumption 2, we have $\|\hat{x}_t - \mathcal{C}\bar{z}_t\|_{\ell_2}^2 \leq C_\rho^2 \rho^{2L} \|\Sigma[\hat{x}_t]\| (n + \log(1/\delta))$ with probability at least $1 - \delta$.*

The proof of Proposition 1 is straightforward, and is deferred to Appendix E. Note that $\|\Sigma[\hat{x}_t]\|$ can grow polynomially in $t$ due to marginal stability. However, we can show that there exists a $\beta > 0$, such that choosing $L \gtrsim \beta \log(T)$, the upper bound in Proposition 1 can be made as small as we want (Ziemann et al., 2023). Another important observation is that, while the true Kalman filter algorithm has a small number of parameters and small memory footprint, the auto-regressive approximation has a larger number of parameters and requires a large memory. Hence, the computational properties of our model differs from that of true Kalman filtering. Nonetheless, its input-output behavior is similar to that of the Kalman filter.

We now further show how to write the $H$ future outputs $y_{t:t+H-1}$ in terms of the covariate $\bar{z}_t$.

$$y_{t:t+H-1} = \mathcal{O}\hat{x}_t + \xi_t = \mathcal{O}\mathcal{C}\bar{z}_t + \mathcal{O}\bar{A}^L\hat{x}_{t-L} + \xi_t, \quad (4.2)$$

where $\xi_t := \mathcal{T}_u u_{t:t+H-2} + \mathcal{T}_e e_{t:t+H-1}$ maps future inputs and future innovations to the future outputs, and is treated as a noise term when predicting $y_{t:t+H-1}$ from $\bar{z}_t$. The matrices $\mathcal{T}_u \in \mathbb{R}^{mH \times p(H-1)}$ and $\mathcal{T}_e \in \mathbb{R}^{mH \times mH}$ denote input Toeplitz matrix and innovation Toeplitz matrix, respectively.

$$\mathcal{T}_u := \begin{bmatrix} 0 & 0 & 0 & \cdots & 0 \\ CB & 0 & 0 & \cdots & 0 \\ CAB & CB & 0 & \cdots & 0 \\ \vdots & \vdots & \vdots & \ddots & \vdots \\ CA^{H-2}B & CA^{H-3}B & CA^{H-4}B & \cdots & CB \end{bmatrix}$$

$$\mathcal{T}_e := \begin{bmatrix} I & 0 & 0 & \cdots & 0 \\ CF & I & 0 & \cdots & 0 \\ CAF & CF & I & \cdots & 0 \\ \vdots & \vdots & \vdots & \ddots & \vdots \\ CA^{H-2}F & CA^{H-3}F & CA^{H-4}F & \cdots & I \end{bmatrix}.$$

The matrix $\mathcal{O} \in \mathbb{R}^{mH \times n}$ is the extended observability matrix, defined as

$$\mathcal{O} := \begin{bmatrix} C \\ CA \\ \vdots \\ CA^{H-1} \end{bmatrix}. \quad (4.3)$$

This matrix maps the latent state to future outputs. Taken together, (4.1) and (4.2) justify the approximation $\hat{x}_t \approx \mathcal{C}\bar{z}_t$ and $y_{t:t+H-1} \approx \mathcal{O}\mathcal{C}\bar{z}_t$, revealing an underlying structure very similar to that posited by our auto-regressive model.

## 4.2. Optimization Results

In this section, we establish, that for any fixed $h \leq r$, the optimization problem $\min_{(G_1, G_2) \in \mathcal{G}(h)} \mathcal{L}_\mathcal{R}(G_1, G_2)$, despite being non-convex, has a nice structure favorable for gradient descent. When the input-output samples $\{(u_t, y_t)\}_{t=0}^{T+H}$ are generated according to (3.1), the loss landscape of $\mathcal{L}_\mathcal{R}$ possesses properties that are often used to establish global convergence of gradient descent. In order to make this precise, we state the following definition.

**Definition 2.** *$X$ is a* local minimum *of $\mathcal{L}$ if $\nabla\mathcal{L}(X) = 0$ and $\lambda_{\min}(\nabla^2\mathcal{L}(X)) \geq 0$. $X$ is a* critical point *of $\mathcal{L}$ if $\nabla\mathcal{L}(X) = 0$. $X$ is a* strict-saddle *point if in addition $\lambda_{\min}(\nabla^2\mathcal{L}(X)) < 0$.*

The proposition below makes this precise.

**Proposition 2.** *Let $\delta \in (0,1)$. Suppose Assumptions 1 and 2 hold. Additionally, suppose we choose $L \gtrsim \beta \log \left( C_\rho T \|C\| \sqrt{C_{\bar{z}} C_{\hat{x}}} \right)/(1 - \rho)$ for a scalar $\beta > 0$, where we set $C_{\bar{z}} := d_{\bar{z}}(\|C\Sigma[\hat{x}_T]C^\top + \Sigma_e\| + \|\Sigma_u\|)$, and $C_{\hat{x}} := n\|\Sigma[\hat{x}_T]\|$. Suppose the trajectory length satisfies,*

$$T \gtrsim L \left( d_{\bar{z}} \log \left( \frac{2T \|\Sigma[\bar{z}_T]\|}{3L\lambda_{\min}(\Sigma[\bar{z}_L])} \right) + \log(1/\delta) \right).$$

*Then, the loss landscape of $\mathcal{L}_\mathcal{R}$ satisfies, with probability at least $1 - \delta$: (i) any local minimum is a global minimum; (ii) any saddle point is a strict-saddle point.*

The proof of Proposition 2 can be found in Appendix F. It follows from Theorem 3 of (Zhu et al., 2020), and certain persistence of excitation properties of the input-output training data $\{(u_t, y_t)\}_{t=0}^{T+H}$.

In general, when a loss function is smooth and satisfies the properties $(i)$ and $(ii)$ stated in Proposition 2, then (perturbed) gradient descent with random initialization is guaranteed to converge to a global minimum (Ge et al., 2015; Lee et al., 2016; Jin et al., 2017). This convergence is only shown to be guaranteed for unconstrained problems which is not the case for our problem. Extending such a result to constrained problems (e.g., via perturbed projected gradient descent) remains an open problem that we leave for future work. Nonetheless, our empirical results in Section 5 suggest that first order optimization methods perform well on our learning objective in practice.

## 4.3. Statistical Results

In this section, we will present our key statistical results. Recall from Section 4.1 that the covariates $\bar{z}_t$ can be approximately mapped to the outputs $y_{t:t+H-1}$ via the Hankel matrix $\mathcal{H} := \mathcal{O}\mathcal{C}$. Our first key result shows that the in-sample prediction error of our model is bounded at global optima of the training loss.

**Theorem 2** (In-sample prediction error)**.** *Let $(\hat{G}_1, \hat{G}_2)$ be the global minimizer of the ERM problem* (3.5)*. Suppose Assumptions 1, 2 hold. Let $\Sigma[\hat{x}_t] := \sum_{k=0}^{t-1} A^k B \Sigma_u B^\top (A^\top)^k + \sum_{k=0}^{t-1} A^k F \Sigma_e F^\top (A^\top)^k$ denote the covariance of the predicted state $\hat{x}_t$, and let $\Sigma[\xi_t] := \mathcal{T}_u(\Sigma_u \otimes I)\mathcal{T}_u^\top + \mathcal{T}_e(\Sigma_e \otimes I)\mathcal{T}_e^\top$ denote the covariance of the offset term $\xi_t$. Suppose, $\max\{\|\mathcal{O}\|_F^2, \|\mathcal{C}\|_F^2\} \leq c_0$. Choose the history length $L \gtrsim L_1$, where*

$$L_1 := \beta \log\left(C_\rho^2 T \|\mathcal{O}\| \|\Sigma[\hat{x}_T]\| / \|\Sigma[\xi_1]\|\right) / (1 - \rho),$$

*for a scalar $\beta > 0$. Define $\Lambda := c_0 C_{\bar{z}} C_\xi$, where $C_\xi := d_{\bar{y}} \|\Sigma[\xi_1]\|$, and $C_{\bar{z}} := d_{\bar{z}}(\|C\Sigma[\hat{x}_T]C^\top + \Sigma_e\| + \|\Sigma_u\|)$. Then, on the training data $\{(\bar{z}_t, y_{t:t+H-1})\}_{t=1}^T$, with probability at least $1 - \delta$, we have*

$$\frac{1}{T} \sum_{t=1}^T \left\|\left(\hat{G}_2\hat{G}_1 - \mathcal{OC}\right)\bar{z}_t\right\|_{\ell_2}^2 \qquad (4.4)$$

$$\lesssim \frac{\|\Sigma[\xi_1]\| H}{T}\left(r(d_{\bar{y}} + d_{\bar{z}})\log(T\Lambda) + \log\left(\frac{T}{\delta}\right)\right).$$

The proof of Theorem 2 is deferred to Appendix B. Since the solution to the ERM problem (3.5) does not have a closed form, we upper bounded the in-sample prediction error in (4.4) with the supremum of the stochastic process $\sum_{t=1}^T 4\langle\xi_t, (G_2G_1 - \mathcal{OC})\bar{z}_t\rangle - \sum_{t=1}^T \|(G_2G_1 - \mathcal{OC})\bar{z}_t\|_{\ell_2}^2$ over the function class $\mathcal{G}$. This can be viewed as a self-normalized version of the Gaussian complexity of $\mathcal{G} - \{(\mathcal{C}, \mathcal{O})\}$ (Ziemann et al., 2022). This technique is crucial for the analysis of in-sample prediction error, in particular, when dealing with the challenges of correlated data and structured function classes. Note that, we get near-optimal dependence on the trajectory length $T$, and the dimensionality $r(d_{\bar{y}} + d_{\bar{z}})$ of the function class $\mathcal{G}(h)$ in (3.4).

Theorem 2 shows that, in the case of unknown dynamics, the prediction from our trained model $\hat{y}_{t:t+H-1} = \hat{G}_2\hat{G}_1\bar{z}_t$ is close to the output prediction $\tilde{y}_{t:t+H-1} = \mathcal{OC}\bar{z}_t$ with known dynamics. Combining this with the filter approximation result, we get $\|\hat{y}_{t:t+H-1} - y_{t:t+H-1}\|_{\ell_2}^2 \leq \tilde{\mathcal{O}}(1/T)$. Note that Theorem 2 gives prediction error bound over the training data. In order to generalize to unseen data, we seek to bound the parameter error. To do so, we need to show that the covariates $\bar{z}_t$ are able to persistently excite all the modes of the Hankel matrix $\mathcal{H} = \mathcal{OC}$. In other words, we require the training data to satisfy $\lambda_{\min}\left(\sum_{t=1}^T \bar{z}_t \bar{z}_t^\top\right) \geq \tilde{\mathcal{O}}(\lambda_{\min}(\Sigma[\bar{z}_L])T)$. Our next result states that it indeed holds under Assumptions 1, 2, and we get near-optimal generalization guarantee.

**Theorem 3** (Parameter Estimation Error)**.** *Consider the same setting of Theorem 2. Additionally, choose the history*

length $L \gtrsim \max\{L_1, L_2\}$, where

$$L_2 := \beta \log\left(C_\rho T \|C\| \sqrt{C_{\bar{z}} C_{\hat{x}}}\right) / (1 - \rho),$$

*for a scalar $\beta > 0$. Suppose the trajectory length satisfies,*

$$T \gtrsim L\left(d_{\bar{z}} \log\left(\frac{2T\|\Sigma[\bar{z}_T]\|}{3L\lambda_{\min}(\Sigma[\bar{z}_L])}\right) + \log(1/\delta)\right).$$

*Then, with probability at least $1 - \delta$, we have*

$$\left\|\hat{G}_2\hat{G}_1 - \mathcal{OC}\right\|_F^2 \lesssim \frac{\|\Sigma[\xi_1]\| H}{\lambda_{\min}(\Sigma[\bar{z}_L]) T}$$
$$\left(r(d_{\bar{y}} + d_{\bar{z}})\log(T\Lambda) + \log\left(\frac{T}{\delta}\right)\right),$$

The proof of Theorem 3 is deferred to Appendix D. Note that Assumptions 1, 2 also guarantee that $\lambda_{\min}(\Sigma[\bar{z}_L]) \succ 0$ (see Theorem 5.4 in Ziemann et al. (2023)), and its lower bound can be estimated in terms of $\lambda_{\min}(\Sigma_u), \lambda_{\min}(\Sigma_v), \mathcal{T}_u, \mathcal{T}_e$ etc. Note that the term $\Lambda$ contains $\|\Sigma[\hat{x}_T]\|$, which can grow polynomially in $T$ when the system (3.1) is marginally stable. However, this does not degrade our bound as $\Lambda$ appears inside logarithm in our results. Also, note that the term $\|\Sigma[\xi_1]\|$ is fixed and does not grow with $T$.

### 4.4. Latent Recovery Result

Finally, we are ready to present the formal statement of our main result previewed in Theorem 1. So far, the parameter error bound only guarantees generalization in terms of model outputs. Now, we show that it furthermore implies latent state estimation, up to similarity transform.

**Theorem 4** (Latent state recovery)**.** *Consider the same settings of Theorems 2, and 3. Additionally, assume that the extended observability matrix $\mathcal{O}$ has full column rank, and the extended controllability matrix $\mathcal{C}$ has full row rank. Suppose the robustness condition holds, that is, we have $2\left\|\hat{G}_2\hat{G}_1 - \mathcal{OC}\right\|_F \leq \min\{\sigma_n(\mathcal{OO}^\top), \sigma_n(\mathcal{CC}^\top)\} =: \sigma_n$. For any new sequence of observed inputs and outputs $\{(u_\tau, y_\tau)\}_{\tau=0}^{t-1}$, let $\hat{x}_t$ be the Kalman filter estimate, and construct the covariate $\bar{z}_t$. Choose $L \gtrsim \max\{L_1, L_2, L_3\}$, where*

$$L_3 := \frac{\beta \log\left(\lambda_{\min}(\Sigma[\bar{z}_L]) T C_\rho^2 \|\hat{x}_t\|_{\ell_2}^2 \sigma_n / \|\bar{z}_t\|_{\ell_2}^2\right)}{1 - \rho}.$$

*Then, there is a similarity transform $S$ such that, with probability at least $1 - \delta$, we have*

$$\left\|\hat{x}_t - S\hat{G}_1\bar{z}_t\right\|_{\ell_2}^2 \lesssim \frac{\|\Sigma[\xi_1]\| H}{\lambda_{\min}(\Sigma[\bar{z}_L]) \sigma_n T}\|\bar{z}_t\|_{\ell_2}^2$$
$$\left(r(d_{\bar{y}} + d_{\bar{z}})\log(T\Lambda) + \log\left(\frac{T}{\delta}\right)\right).$$

The proof of Theorem 4 is presented in Appendix E. We remark that the rank conditions on $\mathcal{O}$ and $\mathcal{C}$ are implied by Assumption 2 as long as $H \geq n$ and $L \geq n$, respectively. Note that, if we choose $H$ to be smaller than $n$, the extended observability matrix $\mathcal{O}$ in (4.3) does not have full column rank, and we might not be able to recover the latent state. On the other hand, if we increase $H$ beyond $n$, the error bound in Theorem 4 becomes loose due to polynomial growth of $\|\Sigma[\xi_1]\|$ in $H$. This shows a trade-off between the length of the predicted outputs and the accuracy latent state recovery. Theorem 4 implies that when $\mathcal{O}$ and $\mathcal{C}$ are full rank, and the size of training data $T$ is sufficiently large (such that $2 \left\| \hat{G}_2 \hat{G}_1 - \mathcal{O}\mathcal{C} \right\|_F \leq \sigma_n$ holds), the weights of our trained auto-regressive model $(\hat{G}_1, \hat{G}_2)$ are close to $(\mathcal{C}, \mathcal{O})$ up to a similarity transform. More specifically, we can combine Theorems 3 and 4 to get the robustness condition in terms of a requirement on the trajectory length $T$ as follows: $T \gtrsim 4\|\Sigma[\xi_1]\| H(r(d_{\bar{y}} + d_{\bar{z}}) \log(T\Lambda) + \log(T/\delta)) / (\lambda_{\min}(\Sigma[\bar{z}_L])\sigma_n^2)$. We remark that, such robustness guarantees are established for classical approaches like the Ho-Kalman factorization (Oymak & Ozay, 2019), which involves performing an SVD of the Hankel matrix. Our auto-regressive model, on the other hand, does not require any additional procedure to guarantee robustness.

# 5. Numerical Experiments

## 5.1. Experimental Setup

We evaluate whether a two-layer linear neural network can learn a latent state. We consider two types of linear dynamical systems: randomly generated synthetic systems and benchmark systems from ControlGym (Zhang et al., 2024).

For the synthetic experiments, we generate data from the system (3.1) with $n = 4$, $p = 2$, and $m = 3$. The dynamics matrix $A$ is constructed by sampling i.i.d. $\mathcal{N}(0, 1)$ entries and rescaled so that $\rho(A) = 1$. The matrices $B$ and $C$ are generated with i.i.d $\mathcal{N}(0, 1/p)$ entries and i.i.d. $\mathcal{N}(0, 1/m)$ entries, respectively. We ensure that the system is observable. We set the process and measurement noise covariance to $\Sigma_w = \sigma_w^2 I_n$ and $\Sigma_v = \sigma_v^2 I_m$, respectively, with $\sigma_w^2 = 0.05$ and $\sigma_v^2 = 0.1$.

For the ControlGym experiments, we use two linear dynamical systems from ControlGym (Zhang et al., 2024): the underwater vehicle environment (ID: umv) and the aircraft environment (ID: ac6). The state dimensions are $n = 8$ for umv and $n = 10$ for ac6.

For each system, a single trajectory $\{(u_t, y_t)\}_{t=0}^{T_{\text{train}}+H}$ is sampled from the system where $u_t \overset{i.i.d}{\sim} \mathcal{N}(0, \Sigma_u)$ with $\Sigma_u = \frac{1}{p} I_p$, and we form a training dataset $\{(\bar{z}_t, y_{t:t+H-1})\}_{t=L}^{T_{\text{train}}}$ from the trajectory.

We train a two-layer linear network whose hidden dimension equals to some $h \in \mathbb{Z}_+$, learning the weights $G_1 \in \mathbb{R}^{h \times (m+p)L}$ and $G_2 \in \mathbb{R}^{mH \times h}$ such that

$$y_{t:t+H-1} \approx G_2 G_1 \bar{z}_t. \tag{5.1}$$

The weights are obtained by minimizing mean-squared error over the training trajectory. We optimize this objective using Adam, a standard adaptive first-order gradient method, with learning rate $\eta_t$ and weight decay $10^{-3}$ in lieu of explicit regularization or parameter constraints. Implementation details and pseudocode are provided in Appendix H.

### 5.1.1. ARCHITECTURE SEARCH ON SYNTHETIC SYSTEMS

In this experiment, we find an optimal hidden dimension $h$ that returns the smallest training loss on the synthetic systems. We repeat the training with $N = 10$ independently generated trajectories and report the smallest average of training losses. We choose $L = 10$, $H = 5$, and $T_{\text{train}} = 10^4$. The learning rate $\eta_t$ is initialized at 0.01 and decays exponentially by a factor $\gamma = 0.9$ every epoch.

### 5.1.2. LATENT STATE RECOVERY

In this experiment, we set the hidden dimension $h$ to the state dimension $n$. We evaluate latent state recovery on both synthetic systems and ControlGym systems. We consider multiple values of $L$, $H$, and $T_{\text{train}}$. Let $\hat{G}_1$ and $\hat{G}_2$ be the learned weights. We sample another trajectory $\{(u_t, y_t)\}_{t=0}^{T_{\text{test}}+H}$ applying nonrandom inputs

$$u_t = \frac{c(t+1)}{T_{\text{test}}} \sin\left(\frac{2\pi t}{T_{\text{period}}}\right) \mathbf{1}_p$$

where $\mathbf{1}_p$ denotes the all-ones vector in $\mathbb{R}^p$, and we set $T_{\text{period}} = 50$, $T_{\text{test}} = 10^3$, and $c = 2$. From the trajectory, collect the feature vectors $\bar{z}_t$ and the activations $\{\hat{G}_1 \bar{z}_t\}_{t=L}^{T_{\text{test}}}$ as well as the predicted state estimates $\{\hat{x}_t\}_{t=0}^{T_{\text{test}}+H}$ using the steady-state Kalman filter predictor form. To handle the non-uniqueness of the state space representation, we fit a linear map $\hat{S} \in \mathbb{R}^{n \times n}$ such that

$$\hat{S} = \underset{S \in \mathbb{R}^{n \times n}}{\operatorname{argmin}} \sum_{t=L}^{T_{\text{test}}} \|\hat{x}_t - S\hat{G}_1 \bar{z}_t\|_{\ell_2}^2. \tag{5.2}$$

The learning rate $\eta_t$ is initialized at 0.05 and decays exponentially by a factor $\gamma = 0.9$ every two epochs.

## 5.2. Architecture Search on Synthetic Systems

Table 1 reports the results of a grid search over the hidden dimension $h$ of the two-layer linear model. For each $h \in \{1, \ldots, 10\}$, we trained the model on $N = 10$ independently generated training trajectories with fixed $L = 10$,

$H = 5$, and $T_{\text{train}} = 10^4$. For each trajectory, we record the minimum training loss over epochs, and then report the sample mean and sample standard deviation of these $N = 10$ values.

*Table 1.* Training loss for different hidden dimension of the auto-regressive model over $N = 10$ trials.

| HIDDEN DIM. | TRAINING LOSS (MEAN $\pm$ STD) |
|:---:|:---:|
| 1 | $309.5980 \pm 167.9855$ |
| 2 | $0.6661 \pm 0.0473$ |
| 3 | $0.7204 \pm 0.0565$ |
| 4 | $\mathbf{0.6179 \pm 0.0408}$ |
| 5 | $0.6921 \pm 0.0586$ |
| 6 | $0.6525 \pm 0.0482$ |
| 7 | $0.6212 \pm 0.0434$ |
| 8 | $0.6234 \pm 0.0453$ |
| 9 | $0.6372 \pm 0.0452$ |
| 10 | $0.6456 \pm 0.0518$ |

The results show that $h = 1$ is under-parameterized, yielding larger loss than the others. Among the grid search values, $h = 4$ attains the lowest average training loss, which is equal to the true state dimension $n = 4$.

### 5.3. Latent State Recovery

We first evaluate latent state recovery on the randomly generated synthetic systems. In order to check whether the learned activations $\hat{S}\hat{G}_1\bar{z}_t$ are aligned with the Kalman filter-based predicted state estimates $\hat{x}_t$, we plot the first coordinate of $\hat{x}_t$ against the corresponding coordinate of $\hat{S}\hat{G}_1\bar{z}_t$ across time $t$. Here, we use $L = 10$ and $T_{\text{train}} = 10^4$ for both $H = 1$ and $H = 5$. The alignment between the two states is illustrated as scatter plots in Figure 2.

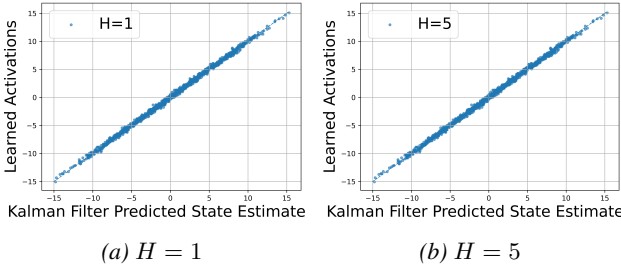

*(a) $H = 1$*      *(b) $H = 5$*

*Figure 2.* Alignment between the predicted state estimates and learned activations. We plot the first coordinate of the predicted state $\hat{x}_t$ against the first coordinate of the learned state $\hat{S}\hat{G}_1\bar{z}_t$, respectively. Each point corresponds to one time index.

Figure 2 shows that the points concentrate tightly around the linear line which implies strong alignment between two values. Similar behavior is observed for the other coordinates (not shown), suggesting that the auto-regressive model can learn the latent states.

We also evaluate latent state recovery using coordinate-wise

$R^2$ statistics. This gives a complementary measure of alignment between the Kalman filter state estimates and the linearly transformed learned activations. Table 2 reports the coordinate-wise and mean $R^2$ statistics for the randomly generated synthetic system and the ControlGym systems.

For the synthetic system, the $R^2$ results are consistent with the scatter plots in Figure 2. The learned activations achieve very high alignment with the Kalman filter state estimates, with mean $R^2 = 0.999$ for $H = 1$ and mean $R^2 = 0.998$ for $H = 5$. All coordinate-wise $R^2$ values are above $0.995$, further supporting that the hidden representation recovers the latent state estimate up to a linear transformation.

To test whether this latent state recovery phenomenon extends beyond randomly generated systems, we repeat the same procedure on the ControlGym systems described in Section 5.1. For these systems, the learned activations also remain strongly aligned with the Kalman filter state estimates. For the underwater vehicle system `umv`, the learned activations achieve mean $R^2 = 0.995$ when $H = 1$ and mean $R^2 = 0.994$ when $H = 5$. For the aircraft system `ac6`, the learned activations achieve mean $R^2 = 0.979$ when $H = 1$ and mean $R^2 = 0.980$ when $H = 5$. Most state coordinates have $R^2$ values close to one. The tenth coordinate of the aircraft system has the lowest $R^2$ values, which are $0.821$ for $H = 1$ and $0.836$ for $H = 5$, but the average alignment remains high.

These results show that the learned activations remain strongly aligned with Kalman filter state estimates on both randomly generated synthetic systems and ControlGym systems. Thus, the latent state recovery phenomenon observed in the synthetic experiments is not limited to the specific random construction, but also appears in more realistic benchmark dynamical systems.

### 5.4. Sample Complexity of Latent State Recovery on Synthetic Systems

We next verify that the model learns better as the amount of training data increases. Figure 3 shows the average $\ell_2$ error between the Kalman filter predictor state $\hat{x}_t$ and the transformed learned activation $\hat{S}\hat{G}_1\bar{z}_t$ as $T_{\text{train}}$ increases, for different history lengths $L$ with fixed future window length $H$.

The recovery error decreases as $T_{\text{train}}$ increases for both $H = 1$ and $H = 5$. This qualitative trend is consistent with the finite-sample latent recovery guarantee in Theorem 4: as the training trajectory becomes longer, the bound on the latent recovery error decreases. While the experiment is not intended as a direct rate verification, it empirically supports the predicted sample-complexity behavior.

*Table 2.* Latent state recovery results on the randomly generated synthetic system and ControlGym systems. We report the coordinate-wise and mean $R^2$ statistics between the Kalman filter predicted state estimates and the linearly transformed learned activations.

| SYSTEM | $H$ | $s_1$ | $s_2$ | $s_3$ | $s_4$ | $s_5$ | $s_6$ | $s_7$ | $s_8$ | $s_9$ | $s_{10}$ | MEAN $R^2$ |
|---|---|---|---|---|---|---|---|---|---|---|---|---|
| RANDOM | 1 | 0.998 | 0.998 | 0.999 | 0.999 | – | – | – | – | – | – | 0.999 |
| RANDOM | 5 | 0.996 | 0.998 | 1.000 | 0.999 | – | – | – | – | – | – | 0.998 |
| UMV | 1 | 1.000 | 0.981 | 1.000 | 1.000 | 0.980 | 1.000 | 1.000 | 1.000 | – | – | 0.995 |
| UMV | 5 | 1.000 | 0.974 | 1.000 | 1.000 | 0.974 | 1.000 | 1.000 | 1.000 | – | – | 0.994 |
| AC6 | 1 | 1.000 | 0.992 | 0.996 | 0.992 | 0.997 | 0.997 | 0.998 | 0.999 | 0.998 | 0.821 | 0.979 |
| AC6 | 5 | 1.000 | 0.991 | 0.996 | 0.991 | 0.996 | 0.995 | 0.997 | 0.998 | 0.997 | 0.836 | 0.980 |

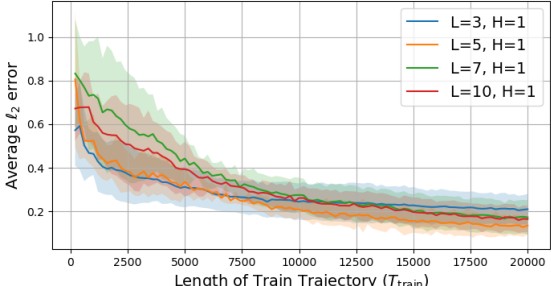

*(a)* Sample complexity plot by varying the history length $L$ with fixed future window length $H = 1$.

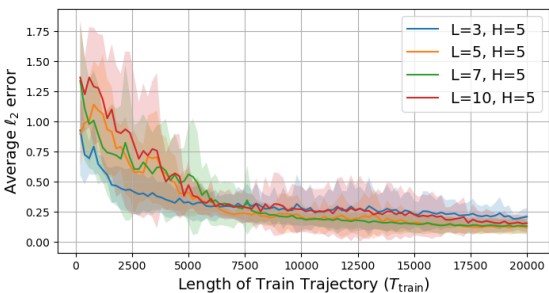

*(b)* Sample complexity plot by varying the history length $L$ with fixed future window length $H = 5$.

*Figure 3.* Sample complexity of latent state recovery. Average $\ell_2$ error between $\hat{x}_t$ and $\hat{S}\hat{G}_1\bar{z}_t$ from the test trajectory versus the length of training trajectory $T_{\text{train}}$ for different history length $L$ with fixed future window length $H$. The lines and the shaded regions indicate the sample mean and $\pm 1$ sample standard deviation, respectively, over 5 trials.

## 6. Conclusion & Discussion

In this work, we theoretically characterized when two-layer linear auto-regressive models naturally learn to approximate Kalman filtering. In particular, we show that the learned hidden representation coincides, up to a similarity transformation, with the state estimates produced by the optimal (Kalman) filter. This occurs simply due to training by empirical risk minimization on a single trajectory, where the model has no explicit knowledge of the underlying dynamics or state.

There are several interesting direction for future work. One is to consider partially observed linear systems driven by non-Gaussian noise. Though the Kalman filter is no longer the optimal estimator, it remains the best linear filter. However, this leads to correlated filter errors, which complicates the statistical analysis. This issue appears in the "shallow" setting as well, and has been noted by Ghai et al. (2020).

Another broad direction is to investigate under which deep learning paradigms latent states naturally emerge for linear systems. For example, recurrent architectures (Hardt et al., 2018) or over-parametrized (deeper) networks. Finally, an impressive capability of LLMs is "in-context learning", a form of meta-learning wherein the model specializes to patterns present in the model inputs. In the context of this work, this would correspond to training a high capacity model on data from many distinct linear dynamical systems, and showing that the model could generalize (and predict latent states) for new systems given sufficiently long input sequences.

## Acknowledgments

The authors are grateful to the anonymous reviewers for their time and effort in reviewing this manuscript and providing thoughtful feedback. S.D. was partly supported by NSF CCF 2312774, NSF OAC-2311521, NSF IIS-2442137, a gift to the LinkedIn-Cornell Bowers CIS Strategic Partnership, and an AI2050 Early Career Fellowship program at Schmidt Sciences. M.F. was supported in part by awards NSF TRIPODS II 2023166, NSF CCF 2212261, NSF CCF 2312775, and by the Moorthy Family professorship at UW.

## Impact Statement

This paper presents work whose goal is to advance the field of Machine Learning. There are many potential societal consequences of our work, none which we feel must be specifically highlighted here.

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

## A. Preliminaries

Expanding the Kalman Filtering predictor form (3.2), we can express $H$ future outputs $y_{t:t+H-1}$ in terms of past inputs and output $\bar{z}_t$, defined in (3.3), as follows,

$$y_{t:t+H-1} = \mathcal{OC}\bar{z}_t + \mathcal{O}\bar{A}^L \hat{x}_{t-L} + \mathcal{T}_u u_{t:t+H-2} + \mathcal{T}_e e_{t:t+H-1}, \tag{A.1}$$

where $\mathcal{O} \in \mathbb{R}^{mH \times n}$ is the observability matrix, $\mathcal{C} \in \mathbb{R}^{n \times (m+p)L}$ is the closed-loop controllability matrix, $\mathcal{T}_u \in \mathbb{R}^{mH \times p(H-1)}$ is a Toeplitz matrix associated with future inputs, and $\mathcal{T}_e \in \mathbb{R}^{mH \times mH}$ is a Toeplitz matrix associated with future innovations, given by

$$\mathcal{O} := \begin{bmatrix} C \\ CA \\ \vdots \\ CA^{H-1} \end{bmatrix}, \qquad \mathcal{C} := \begin{bmatrix} F & \bar{A}F & \cdots & \bar{A}^{L-1}F & B & \bar{A}B & \cdots & \bar{A}^{L-1}B \end{bmatrix}, \tag{A.2}$$

$$\mathcal{T}_u := \begin{bmatrix} 0 & 0 & 0 & \cdots & 0 \\ CB & 0 & 0 & \cdots & 0 \\ CAB & CB & 0 & \cdots & 0 \\ \vdots & \vdots & \vdots & \ddots & \vdots \\ CA^{H-2}B & CA^{H-3}B & CA^{H-4}B & \cdots & CB \end{bmatrix}, \tag{A.3}$$

$$\mathcal{T}_e := \begin{bmatrix} I & 0 & 0 & \cdots & 0 \\ CF & I & 0 & \cdots & 0 \\ CAF & CF & I & \cdots & 0 \\ \vdots & \vdots & \vdots & \ddots & \vdots \\ CA^{H-2}F & CA^{H-3}F & CA^{H-4}F & \cdots & I \end{bmatrix}. \tag{A.4}$$

With these definitions, given the input-output samples $\{(u_t, y_t)\}_{t=0}^{T+H}$ generated from a single trajectory of (3.1), we want to learn an auto-regressive model by solving the following empirical risk minimization (ERM) problem,

$$\hat{n}, \hat{G}_1, \hat{G}_2 = \arg \min_{h \leq r, (G_1, G_2) \in \mathcal{G}(h)} \frac{1}{2T} \sum_{t=1}^{T} \|y_{t:t+H-1} - G_2 G_1 \bar{z}_t\|_{\ell_2}^2. \tag{A.5}$$

## B. In-Sample Prediction Error (Proof of Theorem 2)

**Theorem 2** (Detailed version). *Suppose* $(\hat{G}_1, \hat{G}_2)$ *is the global minimizer of the ERM problem* (A.5). *Suppose Assumptions 1, 2 hold. Let* $\Sigma[\hat{x}_t] := \mathbb{E}\left[\hat{x}_t \hat{x}_t^\top\right] = \sum_{k=0}^{t-1} A^k B \Sigma_u B^\top (A^\top)^k + \sum_{k=0}^{t-1} A^k F \Sigma_e F^\top (A^\top)^k$ *denote the covariance of the predicted state* $\hat{x}_t$, *and let* $\Sigma[\xi_t] := \mathbb{E}[\xi_t \xi_t^\top] = \mathcal{T}_u (\Sigma_u \otimes I) \mathcal{T}_u^\top + \mathcal{T}_e (\Sigma_e \otimes I) \mathcal{T}_e^\top$ *denote the covariance of the offset term* $\xi_t$. *Suppose,* $\max\{\|\mathcal{O}\|_F^2, \|\mathcal{C}\|_F^2\} \leq c_0$, *and choose the history length* $L \geq L_1 := \beta \log \left(C_\rho^2 T \|\mathcal{O}\| \|\Sigma[\hat{x}_T]\| / \|\Sigma[\xi_1]\|\right) / (1 - \rho)$. *Define,*

$$\begin{aligned} C_\xi(\delta) &:= \|\Sigma[\xi_1]\| \left(mH + \log(2T/\delta)\right), \\ C_{\bar{z}}(\delta) &:= \left(\|C\Sigma[\hat{x}_T]C^\top + \Sigma_e\| + \|\Sigma_u\|\right) \left((m+p)L + \log(2T/\delta)\right), \\ \Lambda(\delta) &:= \left(6\sqrt{C_\xi(\delta) C_{\bar{z}}(\delta)}T + 2c_0 C_{\bar{z}}(\delta)T\right) / (H\|\Sigma[\xi_1]\|) \end{aligned} \tag{B.1}$$

*Then, with probability at least* $1 - \delta$, *we have*

$$\frac{1}{T} \sum_{t=1}^{T} \left\|\left(\hat{G}_2 \hat{G}_1 - \mathcal{OC}\right) \bar{z}_t\right\|_{\ell_2}^2 \lesssim \frac{\|\Sigma[\xi_1]\|H}{T} \left(r\left(mH + (m+p)L\right) \log\left(C_0 \Lambda(\delta)\right) + \log\left(\frac{T}{\delta}\right)\right) \tag{B.2}$$

### B.1. Prediction error decomposition

To begin, using the global optimality of $\hat{G}_1, \hat{G}_2$, we have

$$\frac{1}{2T} \sum_{t=1}^{T} \left\|y_{t:t+H-1} - \hat{G}_2 \hat{G}_1 \bar{z}_t\right\|_{\ell_2}^2 \leq \frac{1}{2T} \sum_{t=1}^{T} \|y_{t:t+H-1} - \mathcal{OC}\bar{z}_t\|_{\ell_2}^2.$$

Next, from (A.1), we have $y_{t:t+H-1} = \mathcal{O}\mathcal{C}\bar{z}_t + \mathcal{O}\bar{A}^L\hat{x}_{t-L} + \xi_t = \mathcal{O}\mathcal{C}\bar{z}_t + \zeta_t$, where, we define

$$\zeta_t := \mathcal{O}\bar{A}^L\hat{x}_{t-L} + \xi_t, \qquad \xi_t := \mathcal{T}_u u_{t:t+H-2} + \mathcal{T}_e e_{t:t+H-1}, \tag{B.3}$$

denotes the residual/error terms. Using (B.3) along-with the assumption that $\max\{\|\mathcal{O}\|_F^2, \|\mathcal{C}\|_F^2\} \le c_0$, we get,

$$\sum_{t=1}^{T} \left\| \left(\mathcal{O}\mathcal{C} - \hat{G}_2\hat{G}_1\right)\bar{z}_t \right\|_{\ell_2}^2 \le \sum_{t=1}^{T} 2\left\langle \zeta_t, \left(\hat{G}_2\hat{G}_1 - \mathcal{O}\mathcal{C}\right)\bar{z}_t \right\rangle. \tag{B.4}$$

Adding the positive difference to the right of (B.4), we get the following offset inequality,

$$
\begin{aligned}
\sum_{t=1}^{T} \left\| \left(\hat{G}_2\hat{G}_1 - \mathcal{O}\mathcal{C}\right)\bar{z}_t \right\|_{\ell_2}^2 &\le \sum_{t=1}^{T} 4\left\langle \zeta_t, \left(\hat{G}_2\hat{G}_1 - \mathcal{O}\mathcal{C}\right)\bar{z}_t \right\rangle - \sum_{t=1}^{T} \left\| \left(\mathcal{O}\mathcal{C} - \hat{G}_2\hat{G}_1\right)\bar{z}_t \right\|_{\ell_2}^2, \\
&\le \sum_{t=1}^{T} 6\left\langle \zeta_t, \left(\hat{G}_2\hat{G}_1 - \mathcal{O}\mathcal{C}\right)\bar{z}_t \right\rangle - \sum_{t=1}^{T} 2\left\| \left(\mathcal{O}\mathcal{C} - \hat{G}_2\hat{G}_1\right)\bar{z}_t \right\|_{\ell_2}^2, \\
&\le \sup_{\Theta \in \mathcal{G}-\{\Theta^\star\}} \left\{ \sum_{t=1}^{T} 6\langle \xi_t, \Theta\bar{z}_t \rangle - \sum_{t=1}^{T} \|\Theta\bar{z}_t\|_{\ell_2}^2 \right\}, \\
&\quad + \sup_{\Theta \in \mathcal{G}-\{\Theta^\star\}} \left\{ \sum_{t=1}^{T} 6\left\langle \mathcal{O}\bar{A}^L\hat{x}_{t-L}, \Theta\bar{z}_t \right\rangle - \sum_{t=1}^{T} \|\Theta\bar{z}_t\|_{\ell_2}^2 \right\}, \\
&\le \underbrace{\sup_{\Theta \in \bar{\mathcal{G}}} \left\{ \sum_{t=1}^{T} 6\langle \xi_t, \Theta\bar{z}_t \rangle - \sum_{t=1}^{T} \|\Theta\bar{z}_t\|_{\ell_2}^2 \right\}}_{\text{Martingale offset complexity}} \\
&\quad + \underbrace{9 \left\| \left(\sum_{t=1}^{T} \mathcal{O}\bar{A}^L\hat{x}_{t-L}\bar{z}_t^\top\right) \left(\sum_{t=1}^{T} \bar{z}_t\bar{z}_t^\top\right)^{-1/2} \right\|_F^2}_{\text{Truncation bias}}
\end{aligned} \tag{B.5}
$$

where we get the last inequality by maximizing the second term over all $\Theta \in \mathbb{R}^{mH \times (m+p)L}$, and we define the sets,

$$\mathcal{G} - \{\Theta^\star\} := \{\Theta - \Theta^\star \in \mathbb{R}^{mH \times (m+p)L}; \ \text{rank}(\Theta) \le r; \ \|\Theta\|_F \le c_0\}, \tag{B.6}$$

$$\subseteq \bar{\mathcal{G}} := \{\Theta \in \mathbb{R}^{mH \times (m+p)L}; \ \text{rank}(\Theta) \le 2r; \ \|\Theta\|_F \le 2c_0\}, \tag{B.7}$$

and $\Theta^\star := \mathcal{O}\mathcal{C}$ is the true Hankel matrix. In the following, we will upper bound each term in (B.5) separately to get an upper bound on $(1/T)\sum_{t=1}^{T} \left\| \left(\hat{G}_2\hat{G}_1 - \mathcal{O}\mathcal{C}\right)\bar{z}_t \right\|_{\ell_2}^2$.

## B.2. Martingale offset complexity

**Theorem 5** (Martingale offset complexity). *Under the same setup of Theorem 2, with probability at least $1 - \delta$, we have,*

$$\sup_{\Theta \in \bar{\mathcal{G}}} \left\{ \sum_{t=1}^{T} 6\langle \xi_t, \Theta\bar{z}_t \rangle - \sum_{t=1}^{T} \|\Theta\bar{z}_t\|_{\ell_2}^2 \right\} \le cH\|\Sigma[\xi_1]\| \left( r\left(mH + (m+p)L\right)\log\left(C_0\Lambda(\delta)\right) + \log\left(\frac{1}{\delta}\right) \right), \tag{B.8}$$

*where we define, $\Lambda(\delta) := 6\sqrt{C_\xi(\delta)C_{\bar{z}}(\delta)}T + 2c_0 C_{\bar{z}}(\delta)T / (cH\|\Sigma[\xi_1]\|)$, for some constant $c_0 > 0$.*

### B.2.1. PROOF OF THEOREM 5 (SUPPORTING RESULTS)

We first discretion the set $\bar{\mathcal{G}}$ in (B.7), using $\varepsilon$-covering argument as follows.

**Lemma 1** ($\varepsilon$-covering). *Let $\mathcal{N} := \mathcal{N}(\bar{\mathcal{G}}, \varepsilon, \|\cdot\|_F)$ be an $\varepsilon$-net of $\bar{\mathcal{G}}$ defined in (B.7). Then, with probability at least $1 - \delta$, we have*

$$\sup_{\Theta \in \bar{\mathcal{G}}} \left\{ \sum_{t=1}^{T} \left( 6 \langle \xi_t, \Theta \bar{z}_t \rangle - \|\Theta \bar{z}_t\|_{\ell_2}^2 \right) \right\} \leq \max_{\Theta \in \mathcal{N}} \left\{ \sum_{t=1}^{T} \left( 6 \langle \xi_t, \Theta \bar{z}_t \rangle - \|\Theta \bar{z}_t\|_{\ell_2}^2 \right) \right\},$$
$$+ \varepsilon \left( 6 \sqrt{C_\xi(\delta) C_{\bar{z}}(\delta) T} + 2 c_0 C_{\bar{z}}(\delta) T \right), \tag{B.9}$$

*where $C_\xi(\delta)$, and $C_{\bar{z}}(\delta)$ are as defined in (B.1).*

*Proof.* To begin, denote for all $\Theta \in \bar{\mathcal{G}}$, $X_\Theta$ as

$$X_\Theta := 6 \sum_{t=1}^{T} \xi_t^\top \Theta \bar{z}_t - \sum_{t=1}^{T} \|\Theta \bar{z}_t\|_{\ell_2}^2$$

Then, note that for all $\Theta, \Theta' \in \bar{\mathcal{G}}$, we have

$$|X_\Theta - X_{\Theta'}| \leq \left| 6 \sum_{t=1}^{T} \xi_t^\top \left( \Theta - \Theta' \right) \bar{z}_t \right| + \left| \sum_{t=1}^{T} \bar{z}_t^\top \left( \Theta^\top \Theta - \Theta'^\top \Theta' \right) \bar{z}_t \right|,$$
$$\overset{(i)}{\leq} \|\Theta - \Theta'\|_F \left( 6 \sqrt{\left( \sum_{t=1}^{T} \|\xi_t\|_{\ell_2}^2 \right) \left( \sum_{t=1}^{T} \|\bar{z}_t\|_{\ell_2}^2 \right)} + 2 c_0 \sum_{t=1}^{T} \|\bar{z}_t\|_{\ell_2}^2 \right), \tag{B.10}$$

where we used triangular inequality, followed by Cauchy-Schwarz inequality to obtain (B.10). Finally combining this with Lemma 8, and observing that for every $\Theta \in \bar{\mathcal{G}}$, there exists $\Theta' \in \mathcal{N}(\bar{\mathcal{G}}, \varepsilon, \|\cdot\|_F)$ such that $\|\Theta - \Theta'\|_F \leq \varepsilon$, we get the statement of Lemma 1. $\square$

Next, to get a high probability upper bound on the quantity $\max_{\Theta \in \mathcal{N}} \left\{ \sum_{t=1}^{T} \left( 6 \langle \xi_t, \Theta \bar{z}_t \rangle - \|\Theta \bar{z}_t\|_{\ell_2}^2 \right) \right\}$, we derive the following intermediate result.

**Lemma 2** (Self-normalized offset bound). *Suppose Assumptions 2, 1 hold. Then, for any $\Theta \in \mathcal{N}(\bar{\mathcal{G}}, \varepsilon, \|\cdot\|_F)$, and $\lambda \in \left[ 0, 1/ \left( 18H \| \mathcal{T}_u (\Sigma_u \otimes I) \mathcal{T}_u^\top + \mathcal{T}_e (\Sigma_e \otimes I) \mathcal{T}_e^\top \| \right) \right]$, we have*

$$\mathbb{E} \left[ \exp \left( \lambda \sum_{t=1}^{T} \left( 6 \langle \xi_t, \Theta \bar{z}_t \rangle - \|\Theta \bar{z}_t\|_{\ell_2}^2 \right) \right) \right] \leq 1. \tag{B.11}$$

*Proof.* Recall from (B.3) that $\xi_t = \mathcal{T}_u u_{t:t+H-2} + \mathcal{T}_e e_{t:t+H-1}$. For the ease of notation, and without loss of generality suppose $N := T/H$ is an integer. Then we have

$$\sum_{t=1}^{T} 6 \langle \xi_t, \Theta \bar{z}_t \rangle - \|\Theta \bar{z}_t\|_{\ell_2}^2 = \sum_{\tau=0}^{H-1} \sum_{i=0}^{N-1} 6 \langle \xi_{1+iH+\tau}, \Theta \bar{z}_{1+iH+\tau} \rangle - \|\Theta \bar{z}_{1+iH+\tau}\|_{\ell_2}^2,$$
$$=: \sum_{\tau=0}^{H-1} \sum_{i=0}^{N-1} 6 \langle \xi_{\tau,i}, \Theta \bar{z}_{\tau,i} \rangle - \|\Theta \bar{z}_{\tau,i}\|_{\ell_2}^2, \tag{B.12}$$

where the subscript $(\tau, i)$ denotes the time index $t = iH + \tau + 1$. Applying Hölder's inequality, we have

$$\mathbb{E} \left[ \exp \left( \lambda \sum_{t=1}^{T} 6 \langle \xi_t, \Theta \bar{z}_t \rangle - \|\Theta \bar{z}_t\|_{\ell_2}^2 \right) \right] = \mathbb{E} \left[ \exp \left( \lambda \sum_{\tau=0}^{H-1} \sum_{i=0}^{N-1} 6 \langle \xi_{\tau,i}, \Theta \bar{z}_{\tau,i} \rangle - \|\Theta \bar{z}_{\tau,i}\|_{\ell_2}^2 \right) \right],$$
$$\leq \prod_{\tau=0}^{H-1} \left( \mathbb{E} \left[ \exp \left( \lambda H \sum_{i=0}^{N-1} 6 \langle \xi_{\tau,i}, \Theta \bar{z}_{\tau,i} \rangle - \|\Theta \bar{z}_{\tau,i}\|_{\ell_2}^2 \right) \right] \right)^{1/H}. \tag{B.13}$$

To proceed, let $\{\mathcal{F}_t\}_{t \geq -1}$ be an increasing filtration ($\sigma$-algebra) with all randomness up till time $t-1$. Let $(\tau, i) := iH + \tau + 1$ denote a time index, parameterized by $\tau \in [0, H-1]$, and $i \in [0, N-1]$. Then, we have

$$\mathbb{E}\left[\exp\left(\lambda H \sum_{i=0}^{N-1} 6\langle \xi_{\tau,i}, \Theta\bar{z}_{\tau,i}\rangle - \|\Theta\bar{z}_{\tau,i}\|_{\ell_2}^2\right)\right],$$

$$= \mathbb{E}\left[\exp\left(\lambda H \sum_{i=0}^{N-2} 6\langle \xi_{\tau,i}, \Theta\bar{z}_{\tau,i}\rangle - \|\Theta\bar{z}_{\tau,i}\|_{\ell_2}^2\right)\right.$$
$$\left.\mathbb{E}\left[\exp\left(\lambda H \left(6\langle \xi_{\tau,N-1}, \Theta\bar{z}_{\tau,N-1}\rangle - \|\Theta\bar{z}_{\tau,N-1}\|_{\ell_2}^2\right)\right) | \mathcal{F}_{\tau,N-1}\right]\right],$$

$$\leq \mathbb{E}\left[\exp\left(\lambda H \sum_{i=0}^{N-2} 6\langle \xi_{\tau,i}, \Theta\bar{z}_{\tau,i}\rangle - \|\Theta\bar{z}_{\tau,i}\|_{\ell_2}^2\right)\right], \tag{B.14}$$

where we obtained (B.14) as follows,

$$\mathbb{E}\left[\exp\left(6\lambda H \langle \xi_{\tau,N-1}, \Theta\bar{z}_{\tau,N-1}\rangle - \lambda H \|\Theta\bar{z}_{\tau,N-1}\|_{\ell_2}^2\right) | \mathcal{F}_{\tau,N-1}\right]$$

$$= \exp\left(-\lambda H \|\Theta\bar{z}_{\tau,N-1}\|_{\ell_2}^2\right) \mathbb{E}\left[\exp\left(6\lambda H \langle \xi_{\tau,N-1}, \Theta\bar{z}_{\tau,N-1}\rangle\right) | \mathcal{F}_{\tau,N-1}\right],$$

$$\leq \exp\left(-\lambda H \|\Theta\bar{z}_{\tau,N-1}\|_{\ell_2}^2\right) \exp\left(18\lambda^2 H^2 \left(\|\mathcal{T}_u(\Sigma_u \otimes I)\mathcal{T}_u^\top + \mathcal{T}_e(\Sigma_e \otimes I)\mathcal{T}_e^\top\|\right) \|\Theta\bar{z}_{\tau,N-1}\|_{\ell_2}^2\right),$$

$$= \exp\left(\left(18\lambda^2 H^2 \left(\|\mathcal{T}_u(\Sigma_u \otimes I)\mathcal{T}_u^\top + \mathcal{T}_e(\Sigma_e \otimes I)\mathcal{T}_e^\top\|\right) - \lambda H\right) \|\Theta\bar{z}_{\tau,N-1}\|_{\ell_2}^2\right),$$

$$\leq 1, \tag{B.15}$$

where we obtained the last inequality by choosing $\lambda \in \left[0, 1/\left(18H\|\mathcal{T}_u(\Sigma_u \otimes I)\mathcal{T}_u^\top + \mathcal{T}_e(\Sigma_e \otimes I)\mathcal{T}_e^\top\|\right)\right]$. Repeating the same argument by conditioning on the filtration $\mathcal{F}_{\tau,N-2}, \mathcal{F}_{\tau,N-3}, \ldots, \mathcal{F}_{\tau,0}$ in (B.14), we find that

$$\mathbb{E}\left[\exp\left(\lambda H \sum_{i=0}^{N-1} 6\langle \xi_{\tau,i}, \Theta\bar{z}_{\tau,i}\rangle - \|\Theta\bar{z}_{\tau,i}\|_{\ell_2}^2\right)\right] \leq 1, \tag{B.16}$$

for $\lambda \in \left[0, 1/\left(18H\|\mathcal{T}_u(\Sigma_u \otimes I)\mathcal{T}_u^\top + \mathcal{T}_e(\Sigma_e \otimes I)\mathcal{T}_e^\top\|\right)\right]$. Combining (B.16) with (B.13), we have

$$\mathbb{E}\left[\exp\left(\lambda \sum_{t=1}^{T} 6\langle \xi_t, \Theta\bar{z}_t\rangle - \|\Theta\bar{z}_t\|_{\ell_2}^2\right)\right] \leq \prod_{\tau=0}^{H-1}\left(\mathbb{E}\left[\exp\left(\lambda H \sum_{i=0}^{N-1} 6\langle \xi_{\tau,i}, \Theta\bar{z}_{\tau,i}\rangle - \|\Theta\bar{z}_{\tau,i}\|_{\ell_2}^2\right)\right]\right)^{1/H}$$
$$\leq 1. \tag{B.17}$$

This completes the proof. $\qquad\square$

The result in Lemma 2 immediately leads to the following useful result on the tail of supremum.

**Lemma 3** (Maximal inequality). *Let $\mathcal{N} := \mathcal{N}(\bar{\mathcal{G}}, \varepsilon, \|\cdot\|_F)$ be an $\varepsilon$-net of $\bar{\mathcal{G}}$ defined in (B.7). Let $\Sigma[\xi_t] := \mathbb{E}[\xi_t\xi_t^\top] = \mathcal{T}_u(\Sigma_u \otimes I)\mathcal{T}_u^\top + \mathcal{T}_e(\Sigma_e \otimes I)\mathcal{T}_e^\top$ denote the covariance of $\xi_t$. Then, there exist a universal constant $c > 0$, such that*

$$\mathbb{P}\left(\max_{\Theta \in \mathcal{N}}\left\{\sum_{t=1}^{T}\left(6\langle \xi_t, \Theta\bar{z}_t\rangle - \|\Theta\bar{z}_t\|_{\ell_2}^2\right)\right\} \leq cH\|\Sigma[\xi_1]\|\left(\log\left(|\mathcal{N}|\right) + \log(1/\delta)\right)\right) \geq 1 - \delta \tag{B.18}$$

*Proof.* The proof of lemma 3 uses similar arguments, as used in the proof of Lemma 9 in Ziemann et al. (2022). By jensen's

inequality and monotonicity of the exponential, we have

$$
\exp\left(\lambda\,\mathbb{E}\left[\max_{\Theta\in\mathcal{N}}\left\{\sum_{t=1}^{T}\left(6\left\langle\xi_t,\Theta\bar{z}_t\right\rangle-\|\Theta\bar{z}_t\|_{\ell_2}^2\right)\right\}\right]\right)\leq\mathbb{E}\left[\exp\left(\lambda\max_{\Theta\in\mathcal{N}}\left\{\sum_{t=1}^{T}\left(6\left\langle\xi_t,\Theta\bar{z}_t\right\rangle-\|\Theta\bar{z}_t\|_{\ell_2}^2\right)\right\}\right)\right],
$$

$$
\leq\mathbb{E}\left[\max_{\Theta\in\mathcal{N}}\exp\left(\lambda\left\{\sum_{t=1}^{T}\left(6\left\langle\xi_t,\Theta\bar{z}_t\right\rangle-\|\Theta\bar{z}_t\|_{\ell_2}^2\right)\right\}\right)\right],
$$

$$
\leq\sum_{\Theta\in\mathcal{N}}\mathbb{E}\left[\exp\left(\lambda\left\{\sum_{t=1}^{T}\left(6\left\langle\xi_t,\Theta\bar{z}_t\right\rangle-\|\Theta\bar{z}_t\|_{\ell_2}^2\right)\right\}\right)\right],
$$

$$
\leq\exp\left(\log\left(|\mathcal{N}|\right)\right), \tag{B.19}
$$

where we get the last inequality by choosing $\lambda=1/\left(18H\|\mathcal{T}_u(\Sigma_u\otimes I)\mathcal{T}_u^\top+\mathcal{T}_e(\Sigma_e\otimes I)\mathcal{T}_e^\top\|\right)$, and applying Lemma 2. Combining this with the Chernoff bound, we get

$$
\mathbb{P}\left(\max_{\Theta\in\mathcal{N}}\left\{\sum_{t=1}^{T}\left(6\left\langle\xi_t,\Theta\bar{z}_t\right\rangle-\|\Theta\bar{z}_t\|_{\ell_2}^2\right)\right\}-\frac{1}{\lambda}\log\left(|\mathcal{N}|\right)\geq\kappa\right)
$$

$$
\leq\mathbb{P}\left(\exp\left(\lambda\max_{\Theta\in\mathcal{N}}\left\{\sum_{t=1}^{T}\left(6\left\langle\xi_t,\Theta\bar{z}_t\right\rangle-\|\Theta\bar{z}_t\|_{\ell_2}^2\right)\right\}-\log\left(|\mathcal{N}|\right)\right)\geq\exp\left(\lambda\kappa\right)\right),
$$

$$
\leq\mathbb{E}\left[\exp\left(\lambda\max_{\Theta\in\mathcal{N}}\left\{\sum_{t=1}^{T}\left(6\left\langle\xi_t,\Theta\bar{z}_t\right\rangle-\|\Theta\bar{z}_t\|_{\ell_2}^2\right)\right\}-\log\left(|\mathcal{N}|\right)\right)\right]\exp\left(-\lambda\kappa\right),
$$

$$
\leq\exp\left(-\lambda\kappa\right), \tag{B.20}
$$

where we get the last inequality by choosing $\lambda=1/\left(18H\|\mathcal{T}_u(\Sigma_u\otimes I)\mathcal{T}_u^\top+\mathcal{T}_e(\Sigma_e\otimes I)\mathcal{T}_e^\top\|\right)$, and using (B.19). Finally, choosing $\kappa=\frac{1}{\lambda}\log(1/\delta)$, we get the statement of the lemma. $\qquad\square$

Next, to upper bound the cardinality of the $\varepsilon$-covering set $\mathcal{N}(\bar{\mathcal{G}},\varepsilon,\|\cdot\|_F)$, we use a slightly modified version of Lemma 4.5. in Recht et al. (2010), states as follows,

**Lemma 4** (Cardinality of $\mathcal{N}$). *Let* $\varepsilon\in(0,1)$. *Let* $\mathcal{N}(\bar{\mathcal{G}},\varepsilon,\|\cdot\|_F)$ *be an* $\varepsilon$-*net of* $\bar{\mathcal{G}}$ *with respect to* $\|\cdot\|_F$ *of minimal cardinality. Then, we have*

$$
|\mathcal{N}|\leq\left(\frac{C_0}{\varepsilon}\right)^{r(mH+(m+p)L-2r)} \tag{B.21}
$$

### B.2.2. FINALIZING THE PROOF OF THEOREM 5

To finalize the proof of Theorem 5, we first combine Lemma 3 with Lemma 4 to obtain,

$$
\mathbb{P}\left(\max_{\Theta\in\mathcal{N}}\left\{\sum_{t=1}^{T}\left(6\left\langle\xi_t,\Theta\bar{z}_t\right\rangle-\|\Theta\bar{z}_t\|_{\ell_2}^2\right)\right\}\leq cH\|\Sigma[\xi_1]\|\left(r\left(mH+(m+p)L\right)\log\left(\frac{C_0}{\varepsilon}\right)+\log\left(\frac{1}{\delta}\right)\right)\right)\geq1-\delta,
$$

Combining this with Lemma 1, with probability at least $1-\delta$, we have

$$
\sup_{\Theta\in\bar{\mathcal{G}}}\left\{\sum_{t=1}^{T}\left(6\left\langle\xi_t,\Theta\bar{z}_t\right\rangle-\|\Theta\bar{z}_t\|_{\ell_2}^2\right)\right\}\leq cH\|\Sigma[\xi_1]\|\left(r\left(mH+(m+p)L\right)\log\left(\frac{C_0}{\varepsilon}\right)+\log\left(\frac{1}{\delta}\right)\right)
$$

$$
+\varepsilon\left(6\sqrt{C_\xi(\delta)C_{\bar{z}}(\delta)T}+2c_0C_{\bar{z}}(\delta)T\right). \tag{B.22}
$$

Finally choosing $\varepsilon=\frac{cH\|\Sigma[\xi_1]\|}{6\sqrt{C_\xi(\delta)C_{\bar{z}}(\delta)T}+2c_0C_{\bar{z}}(\delta)T}$, we get the statement of Theorem 5.

## B.3. Truncation bias

**Lemma 5** (Truncation bias)**.** *Suppose Assumptions 1, 2 hold, and we choose the history length* $L \geq \beta \log \left( C_\rho^2 T \|\mathcal{O}\| \|\Sigma[\hat{x}_T]\| / \|\Sigma[\xi_1]\| \right) / (1 - \rho)$. *Then, there exist a universal constant* $c > 0$ *such that, with probability at least* $1 - \delta$, *we have*

$$\left\| \left( \sum_{t=1}^T \mathcal{O} \bar{A}^L \hat{x}_{t-L} \bar{z}_t^\top \right) \left( \sum_{t=1}^T \bar{z}_t \bar{z}_t^\top \right)^{-1/2} \right\|_F^2 \leq c \|\Sigma[\xi_1]\| \left( n + \log(T/\delta) \right), \tag{B.23}$$

*Proof.*

$$\left\| \left( \sum_{t=1}^T \mathcal{O} \bar{A}^L \hat{x}_{t-L} \bar{z}_t^\top \right) \left( \sum_{t=1}^T \bar{z}_t \bar{z}_t^\top \right)^{-1/2} \right\|_F^2 \leq \left\| \mathcal{O} \bar{A}^L \right\|^2 \left\| \left( \sum_{t=1}^T \hat{x}_{t-L} \bar{z}_t^\top \right) \left( \sum_{t=1}^T \bar{z}_t \bar{z}_t^\top \right)^{-1/2} \right\|_F^2,$$

$$\leq \|\mathcal{O}\| C_\rho^2 \rho^{2L} \sum_{t=1}^T \|\hat{x}_{t-L}\|_{\ell_2}^2. \tag{B.24}$$

Hence, using Lemma 8, with probability at least $1 - \delta$, we have

$$\left\| \left( \sum_{t=1}^T \mathcal{O} \bar{A}^L \hat{x}_{t-L} \bar{z}_t^\top \right) \left( \sum_{t=1}^T \bar{z}_t \bar{z}_t^\top \right)^{-1/2} \right\|_F^2 \leq c \|\mathcal{O}\| C_\rho^2 \rho^{2L} T \|\Sigma[\hat{x}_T]\| \left( n + \log(T/\delta) \right),$$

$$\overset{(i)}{\leq} c \|\Sigma[\xi_1]\| \left( n + \log(T/\delta) \right), \tag{B.25}$$

where we obtained (i) by using the argument that, there exists a $\beta > 0$ such that,

$$C_\rho^2 \rho^{2L} T \|\mathcal{O}\| \|\Sigma[\hat{x}_T]\| \leq \|\Sigma[\xi_1]\| \iff L \geq \beta \log \left( C_\rho^2 T \|\mathcal{O}\| \|\Sigma[\hat{x}_T]\| / \|\Sigma[\xi_1]\| \right) / (1 - \rho). \tag{B.26}$$

This completes the proof. $\square$

## B.4. Finalizing the Proof of Theorem 2

To finalize the proof of Theorem 2, we combine Theorem 5, Lemma 5, and (B.5) to get the following result: Choosing $L \geq \beta \log \left( C_\rho^2 T \|\mathcal{O}\| \|\Sigma[\hat{x}_T]\| / \|\Sigma[\xi_1]\| \right) / (1 - \rho)$, with probability at least $1 - \delta$, we have

$$\frac{1}{T} \sum_{t=1}^T \left\| \left( \hat{G}_2 \hat{G}_1 - \mathcal{O}\mathcal{C} \right) \bar{z}_t \right\|_{\ell_2}^2 \leq \frac{cH \|\Sigma[\xi_1]\|}{T} \left( r \left( mH + (m+p)L \right) \log \left( C_0 \Lambda(\delta) \right) + \log \left( \frac{1}{\delta} \right) \right)$$

$$+ \frac{c \|\Sigma[\xi_1]\|}{T} \left( n + \log(T/\delta) \right),$$

$$\leq \frac{cH \|\Sigma[\xi_1]\|}{T} \left( r \left( mH + (m+p)L \right) \log \left( C_0 \Lambda(\delta) \right) + \log \left( \frac{T}{\delta} \right) \right), \tag{B.27}$$

where we obtained the last inequality by the choosing the regularization parameter,

$$\lambda \leq \frac{cH \|\Sigma[\xi_1]\|}{3 c_0 T} \left( r \left( mH + (m+p)L \right) \log \left( C_0 \Lambda(\delta) \right) + \log \left( \frac{T}{\delta} \right) \right). \tag{B.28}$$

This completes the proof of Theorem 2.

# C. Persistence of excitation

**Theorem 6** (Persistence of excitation)**.** *Fix a failure probability* $\delta \in (0, 1)$, *and suppose Assumptions 1, 2 hold. For any* $t \in [1, T]$, *let* $\Sigma[\bar{z}_t] := \mathbb{E}[\bar{z}_t \bar{z}_t^\top]$, *and* $\Sigma[\hat{x}_t] := \mathbb{E}[\hat{x}_t \hat{x}_t^\top]$. *Define,*

$$C_{\hat{x}}(\delta) := \|\Sigma[\hat{x}_T]\| \left( n + \log(2T/\delta) \right), \quad C_{\bar{z}}(\delta) := \left( \|C \Sigma[\hat{x}_T] C^\top + \Sigma_e\| + \|\Sigma_u\| \right) \left( (m+p)L + \log(2T/\delta) \right), \tag{C.1}$$

*and choose $L \gtrsim L_2 := \beta \log\left(C_\rho T \|C\| \sqrt{C_{\bar{z}}(\delta) C_{\hat{x}}(\delta)}/c \log(T)\right)/(1-\rho)$. Suppose the trajectory length satisfies,*

$$T \gtrsim L\left((m+p)L \log\left(\frac{2T \|\Sigma[\bar{z}_T]\|}{3L\lambda_{\min}(\Sigma[\bar{z}_L])}\right) + \log(1/\delta)\right).$$

*Then, we have*

$$\mathbb{P}\left(\sum_{t=1}^{T} \bar{z}_t \bar{z}_t^\top \succeq \frac{T}{18}\Sigma[\bar{z}_L]\right) \geq 1-\delta, \quad and \quad \Sigma[\bar{z}_L] = \mathbb{E}[\bar{z}_L \bar{z}_L^\top] \succ 0. \tag{C.2}$$

*Proof.* The proof of Theorem 6 follows similar arguments (with certain modifications) as used in the proof of Theorem 5.2 in Ziemann et al. (2023). First, recall that $\bar{z}_t = \begin{bmatrix} y_{t-1}^\top & \cdots & y_{t-L}^\top & u_{t-1}^\top & \cdots & u_{t-L}^\top \end{bmatrix}^\top \in \mathbb{R}^{(m+p)L}$. To apply Theorem 5.2 in Ziemann et al. (2023), we need to write the evolution of the covariates $\bar{z}_t$ for $t > 1$ in terms of the covariates of an ARX model (after subtracting the truncation bias at each time step). Using (3.2), we can easily show that

$$\bar{z}_{t+1} = \mathcal{A}\bar{z}_t + \mathcal{B}\bar{v}_t + \bar{b}_t, \tag{C.3}$$

where, we define

$$\mathcal{A} := \begin{bmatrix} \mathcal{A}_{11} & \mathcal{A}_{12} \\ 0_{pL \times mL} & \mathcal{A}_{22} \end{bmatrix} \in \mathbb{R}^{(m+p)L \times (m+p)L}$$

$$\mathcal{A}_{11} := \begin{bmatrix} C F & C\bar{A}F & \cdots & C\bar{A}^{L-2}F & C\bar{A}^{L-1}F \\ I_m & 0 & \cdots & 0 & 0 \\ 0 & I_m & \cdots & 0 & 0 \\ \vdots & \vdots & \ddots & \vdots & \vdots \\ 0 & 0 & \cdots & I_m & 0 \end{bmatrix} \in \mathbb{R}^{mL \times mL}$$

$$\mathcal{A}_{12} := \begin{bmatrix} C B & C\bar{A}B & \cdots & C\bar{A}^{L-2}B & C\bar{A}^{L-1}B \\ 0 & 0 & \cdots & 0 & 0 \\ 0 & 0 & \cdots & 0 & 0 \\ \vdots & \vdots & \ddots & \vdots & \vdots \\ 0 & 0 & \cdots & 0 & 0 \end{bmatrix} \in \mathbb{R}^{mL \times pL}$$

$$\mathcal{A}_{22} := \begin{bmatrix} 0 & 0 & \cdots & 0 & 0 \\ I_p & 0 & \cdots & 0 & 0 \\ 0 & I_p & \cdots & 0 & 0 \\ \vdots & \vdots & \ddots & \vdots & \vdots \\ 0 & 0 & \cdots & I_p & 0 \end{bmatrix} \in \mathbb{R}^{pL \times pL}$$

$$\mathcal{B} := \begin{bmatrix} \Sigma_e^{1/2} & 0 \\ 0 & 0 \\ \vdots & \vdots \\ 0 & \Sigma_u^{1/2} \\ \vdots & \vdots \\ 0 & 0 \end{bmatrix} \in \mathbb{R}^{(m+p)L \times (m+p)}, \quad \bar{b}_t := \begin{bmatrix} C\bar{A}^L \hat{x}_{t-L} \\ 0_{(m+p)L-m} \end{bmatrix} \in \mathbb{R}^{(m+p)L}, \quad \bar{v} := \begin{bmatrix} \Sigma_e^{-1/2} e_t \\ \Sigma_u^{-1/2} u_t \end{bmatrix} \in \mathbb{R}^{m+p}$$

For the matrix $\mathcal{B}$, only the blocks at $(1,1)$ and $(L+1, 2)$ are Identity, and the rest is zero. Note that, Theorem 5.2 in Ziemann et al. (2023) requires $\rho(\mathcal{A}_{11}) \leq 1$, so that the associated ARX model is non-explosive. In our case, due to Assumption 2, we have $\rho(A) \leq 1$ and $\rho(\bar{A}) < 1$. As a result our system (3.1) is non-explosive. Hence, we do not need additional assumption on $\mathcal{A}_{11}$. Expanding (C.3), we have

$$\bar{z}_{t+1} = \mathcal{A}\bar{z}_t + \mathcal{B}\bar{v}_t + \bar{b}_t \implies \bar{z}_t = \sum_{k=0}^{t-1} \mathcal{A}^{t-1-k}(\mathcal{B}\bar{v}_k + \bar{b}_k). \tag{C.4}$$

Hence, the vector $\bar{z}_{1:T}$ of all covariates satisfies the following causal linear relation,

$$
\begin{bmatrix} \bar{z}_1 \\ \bar{z}_2 \\ \bar{z}_3 \\ \vdots \\ \bar{z}_T \end{bmatrix} = \underbrace{\begin{bmatrix} \mathcal{B} & 0 & 0 & \cdots & 0 \\ \mathcal{AB} & \mathcal{B} & 0 & \cdots & 0 \\ \mathcal{A}^2\mathcal{B} & \mathcal{AB} & \mathcal{B} & \cdots & 0 \\ \vdots & \vdots & \vdots & \ddots & \vdots \\ \mathcal{A}^{T-1}\mathcal{B} & \mathcal{A}^{T-2}\mathcal{B} & \mathcal{A}^{T-3}\mathcal{B} & \cdots & \mathcal{B} \end{bmatrix}}_{\text{evolution of excitations}} \begin{bmatrix} \bar{v}_1 \\ \bar{v}_2 \\ \bar{v}_3 \\ \vdots \\ \bar{v}_{T-1} \end{bmatrix} + \underbrace{\begin{bmatrix} I & 0 & 0 & \cdots & 0 \\ \mathcal{A} & I & 0 & \cdots & 0 \\ \mathcal{A}^2 & \mathcal{A} & I & \cdots & 0 \\ \vdots & \vdots & \vdots & \ddots & \vdots \\ \mathcal{A}^{T-1} & \mathcal{A}^{T-2} & \mathcal{A}^{T-3} & \cdots & I \end{bmatrix}}_{\text{evolution of bias}} \begin{bmatrix} \bar{b}_1 \\ \bar{b}_2 \\ \bar{b}_3 \\ \vdots \\ \bar{b}_{T-1} \end{bmatrix} \quad \text{(C.5)}
$$

From (C.5), it is easy to see that,

$$
\begin{aligned}
\bar{z}_t \bar{z}_t^\top &= \sum_{k=0}^{t-1} \sum_{l=0}^{t-1} \mathcal{A}^{t-1-k} (\mathcal{B}\bar{v}_k + \bar{b}_k)(\mathcal{B}\bar{v}_l + \bar{b}_l)^\top (\mathcal{A}^\top)^{t-1-l}, \\
&= \sum_{k=0}^{t-1} \sum_{l=0}^{t-1} \mathcal{A}^{t-1-k} \mathcal{B}\bar{v}_k \bar{v}_l^\top \mathcal{B}^\top (\mathcal{A}^\top)^{t-1-l} + \sum_{k=0}^{t-1} \sum_{l=0}^{t-1} \mathcal{A}^{t-1-k} \mathcal{B}\bar{v}_k \bar{b}_l^\top (\mathcal{A}^\top)^{t-1-l}, \\
&+ \sum_{k=0}^{t-1} \sum_{l=0}^{t-1} \mathcal{A}^{t-1-k} \bar{b}_k \bar{v}_l^\top \mathcal{B}^\top (\mathcal{A}^\top)^{t-1-l} + \sum_{k=0}^{t-1} \sum_{l=0}^{t-1} \mathcal{A}^{t-1-k} \bar{b}_k \bar{b}_l^\top (\mathcal{A}^\top)^{t-1-l}, \\
&\succeq \sum_{k=0}^{t-1} \sum_{l=0}^{t-1} \mathcal{A}^{t-1-k} \mathcal{B}\bar{v}_k \bar{v}_l^\top \mathcal{B}^\top (\mathcal{A}^\top)^{t-1-l} + \sum_{k=0}^{t-1} \sum_{l=0}^{t-1} \mathcal{A}^{t-1-k} \mathcal{B}\bar{v}_k \bar{b}_l^\top (\mathcal{A}^\top)^{t-1-l}, \\
&+ \sum_{k=0}^{t-1} \sum_{l=0}^{t-1} \mathcal{A}^{t-1-k} \bar{b}_k \bar{v}_l^\top \mathcal{B}^\top (\mathcal{A}^\top)^{t-1-l}. \quad \text{(C.6)}
\end{aligned}
$$

Therefore, using Weyl's inequality, we have

$$
\begin{aligned}
\lambda_{\min}\left(\sum_{t=1}^T \bar{z}_t \bar{z}_t^\top\right) &\succeq \lambda_{\min}\left(\sum_{t=1}^T \sum_{k=0}^{t-1} \sum_{l=0}^{t-1} \mathcal{A}^{t-1-k} \mathcal{B}\bar{v}_k \bar{v}_l^\top \mathcal{B}^\top (\mathcal{A}^\top)^{t-1-l}\right) \\
&- \left\| \sum_{t=1}^T \sum_{k=0}^{t-1} \sum_{l=0}^{t-1} 2\mathcal{A}^{t-1-k} \bar{b}_k \bar{v}_l^\top \mathcal{B}^\top (\mathcal{A}^\top)^{t-1-l} \right\|, \\
&=: \lambda_{\min}\left(\sum_{t=1}^T \bar{z}_t' \bar{z}_t'^\top\right) - 2\left\| \sum_{t=1}^T \bar{z}_t' b_t'^\top \right\|, \quad \text{(C.7)}
\end{aligned}
$$

where we define, for all $t \geq 1$

$$
\bar{z}_t' = \bar{z}_t - b_t' = \mathcal{A}\bar{z}_{t-1}' + \mathcal{B}\bar{v}_{t-1}, \quad \text{and} \quad b_t' = \mathcal{A}b_{t-1}' \quad \text{(C.8)}
$$

The first term in (C.7) can be lower bounded by directly applying Theorem 5.2 in Ziemann et al. (2023) with $\tau = L$. To upper bound the second term in (C.7), we use a similar argument as used in the proof of Lemma 5. Using Cauchy-Schwarz inequality along-with Lemma 8, with probability at least $1 - \delta$, we have

$$
\left\| \sum_{t=1}^T \bar{z}_t' b_t'^\top \right\| \leq \sqrt{\left(\sum_{t=1}^T \|\bar{z}_t'\|_{\ell_2}^2\right)\left(\sum_{t=1}^T \|b_t'\|_{\ell_2}^2\right)} \leq \|C\| C_\rho \rho^L T \sqrt{C_{\bar{z}}(\delta) C_{\hat{x}}(\delta)} \leq c \log(T). \quad \text{(C.9)}
$$

where we define,

$$
C_{\hat{x}}(\delta) := \|\Sigma[\hat{x}_T]\| \left(n + \log(2T/\delta)\right), \quad C_{\bar{z}}(\delta) := \left(\|C\Sigma[\hat{x}_T]C^\top + \Sigma_e\| + \|\Sigma_u\|\right)\left((m+p)L + \log(2T/\delta)\right), \quad \text{(C.10)}
$$

and we obtained the last inequality by choosing $L > 0$ such that

$$
\begin{aligned}
\|C\| C_\rho \rho^L T &\sqrt{C_{\bar{z}}(\delta) C_{\hat{x}}(\delta)} \leq c \log(T), \\
&\Longleftarrow L \geq \beta \log\left(C_\rho T \|C\| \sqrt{C_{\bar{z}}(\delta) C_{\hat{x}}(\delta)}/c \log(T)\right)/(1-\rho).
\end{aligned}
$$

Combining (C.9), and the result of Theorem 5.2 applied to upper bound the first term in (C.7), we get the first statement of Theorem 6. The second statement that $\Sigma[\bar{z}_L] \succ 0$ follows readily from combing Assumptions 1, 2 with Theorem 5.4 in Ziemann et al. (2023). This completes the proof. □

## D. Parameter Estimation Error (Proof of Theorem 3)

**Theorem 3** (Detailed version). *Under the setting of Theorems 2, 6, with probability at least $1 - \delta$, we have*

$$\left\| \hat{G}_2 \hat{G}_1 - \mathcal{O}\mathcal{C} \right\|_F^2 \lesssim \frac{\|\Sigma[\xi_1]\| H}{\lambda_{\min}\left(\Sigma[\bar{z}_L]\right) T} \left( r\left(mH + (m+p)L\right) \log\left(C_0 \Lambda(\delta)\right) + \log\left(\frac{T}{\delta}\right) \right). \tag{D.1}$$

*Proof.* Theorem 3 follows directly from combining Theorems 2 and 6. To begin, we observe that,

$$\sum_{t=1}^{T} \left\| \left( \hat{G}_2 \hat{G}_1 - \mathcal{O}\mathcal{C} \right) \bar{z}_t \right\|_{\ell_2}^2 = \sum_{t=1}^{T} \operatorname{tr}\left( \bar{z}_t^T \left( \hat{G}_2 \hat{G}_1 - \mathcal{O}\mathcal{C} \right)^\top \left( \hat{G}_2 \hat{G}_1 - \mathcal{O}\mathcal{C} \right) \bar{z}_t \right),$$

$$= \operatorname{tr}\left( \left( \hat{G}_2 \hat{G}_1 - \mathcal{O}\mathcal{C} \right)^\top \left( \hat{G}_2 \hat{G}_1 - \mathcal{O}\mathcal{C} \right) \sum_{t=1}^{T} \bar{z}_t \bar{z}_t^T \right), \tag{D.2}$$

Combining this with Theorem 2, with probability at least $1 - \delta$, we have

$$\left\| \hat{G}_2 \hat{G}_1 - \mathcal{O}\mathcal{C} \right\|_F^2 \lesssim \frac{\|\Sigma[\xi_1]\| H}{\lambda_{\min}\left( \sum_{t=1}^{T} \bar{z}_t \bar{z}_t^T \right)} \left( r\left(mH + (m+p)L\right) \log\left(C_0 \Lambda(\delta)\right) + \log\left(\frac{T}{\delta}\right) \right), \tag{D.3}$$

which is then combined with Theorem 6 to get the statement of Theorem 3. □

## E. Latent State Recovery (Proof of Theorem 4)

**Theorem 4** (Detailed version). *Consider the same settings of Theorems 2, 6. Additionally, assume that the extended observability matrix $\mathcal{O}$ has full column rank, and the extended controllability matrix $\mathcal{C}$ has full row rank. Suppose the robustness condition $2\|\hat{G}_2\hat{G}_1 - \mathcal{O}\mathcal{C}\| \leq \min\left\{ \sigma_n\left(\mathcal{O}\mathcal{O}^\top\right), \sigma_n\left(\mathcal{C}\mathcal{C}^\top\right) \right\} =: \sigma_n$ holds. For any new sequence of observed inputs-outputs $\{(u_\tau, y_\tau)\}_{\tau=0}^{t-1}$, let $\hat{x}_t$ be the Kalman filter estimate and $\bar{z}_t$ be the constructed covaraite. Suppose we choose $L \gtrsim \max\{L_1, L_2, L_3\}$, where*

$$L_3 := \beta \log\left( \frac{\lambda_{\min}\left(\Sigma[\bar{z}_L]\right) T C_\rho^2 \|\hat{x}_{t-L}\|_{\ell_2}^2 \sigma_n}{\|\Sigma[\xi_1]\| H \left( r\left(mH + (m+p)L\right) \log\left(C_0 \Lambda(\delta)\right) + \log\left(\frac{T}{\delta}\right) \right) \|\bar{z}_t\|_{\ell_2}^2} \right) (1 - \rho)^{-1}.$$

*Then, there is a similarity transform $S$ such that, with probability at least $1 - \delta$, we have*

$$\left\| \hat{x}_t - S\hat{G}_1 \bar{z}_t \right\|_{\ell_2}^2 \lesssim \frac{\|\Sigma[\xi_1]\| H}{\lambda_{\min}\left(\Sigma[\bar{z}_L]\right) \sigma_n T} \left( r\left(mH + (m+p)L\right) \log\left(C_0 \Lambda(\delta)\right) + \log\left(\frac{T}{\delta}\right) \right) \|\bar{z}_t\|_{\ell_2}^2,$$

### E.1. Robustness of Parameter Estimation

Before we state a supporting lemma to prove Theorem 4, we introduce the following notation, to denote the distance between two matrices of appropriate dimensions up to a similarity transform

$$\operatorname{dist}\left(X, \hat{X}\right) := \min_{S \in \mathbb{R}^{n \times n} : S^\top S = I_n} \left\| X - S\hat{X} \right\|_F \tag{E.1}$$

The following lemma is adapted from Tu et al. (2016), and is used to connect parameter estimation guarantee to the latent state recovery guarantee.

**Lemma 6.** *Let $\mathcal{O} \in \mathbb{R}^{mH \times n}$, and $\mathcal{C} \in \mathbb{R}^{n \times (p+m)L}$ be two rank $n$ matrices. Given two matrices $\hat{G}_1, \hat{G}_2$ of appropriate dimensions such that $\|\hat{G}_2\hat{G}_1 - \mathcal{O}\mathcal{C}\| \leq \frac{1}{2} \min\left\{ \sigma_n\left(\mathcal{O}\mathcal{O}^\top\right), \sigma_n\left(\mathcal{C}\mathcal{C}^\top\right) \right\}$. Then, we have*

$$\operatorname{dist}^2\left( \begin{bmatrix} \mathcal{O} \\ \mathcal{C}^\top \end{bmatrix}, \begin{bmatrix} \hat{G}_2 \\ \hat{G}_1^\top \end{bmatrix} \right) \leq \frac{2}{\sqrt{2} - 1} \frac{\left\| \hat{G}_2\hat{G}_1 - \mathcal{O}\mathcal{C} \right\|_F^2}{\min\left\{ \sigma_n\left(\mathcal{O}\mathcal{O}^\top\right), \sigma_n\left(\mathcal{C}\mathcal{C}^\top\right) \right\}} \tag{E.2}$$

*Proof.* We have

$$\begin{bmatrix} 0 & \mathcal{O}\mathcal{C} \\ \mathcal{C}^\top\mathcal{O}^\top & 0 \end{bmatrix} - \begin{bmatrix} 0 & \hat{G}_2\hat{G}_1 \\ \hat{G}_1^\top\hat{G}_2^\top & 0 \end{bmatrix}$$

$$= \frac{1}{2}\begin{bmatrix} \mathcal{O} & \hat{G}_2 \\ \mathcal{C}^\top & -\hat{G}_1^\top \end{bmatrix}\begin{bmatrix} \mathcal{O} & \hat{G}_2 \\ \mathcal{C}^\top & -\hat{G}_1^\top \end{bmatrix}^\top - \frac{1}{2}\begin{bmatrix} \mathcal{O} & \hat{G}_2 \\ -\mathcal{C}^\top & \hat{G}_1^\top \end{bmatrix}\begin{bmatrix} \mathcal{O} & \hat{G}_2 \\ -\mathcal{C}^\top & \hat{G}_1^\top \end{bmatrix}^\top. \tag{E.3}$$

Furthermore,

$$\begin{bmatrix} \mathcal{O} & \hat{G}_2 \\ -\mathcal{C}^\top & \hat{G}_1^\top \end{bmatrix}\begin{bmatrix} \mathcal{O} & \hat{G}_2 \\ -\mathcal{C}^\top & \hat{G}_1^\top \end{bmatrix}^\top = \begin{bmatrix} \mathcal{O}\mathcal{O}^\top + \hat{G}_2\hat{G}_2^\top & -\mathcal{O}\mathcal{C} + \hat{G}_2\hat{G}_1 \\ -\mathcal{C}^\top\mathcal{O}^\top + \hat{G}_1^\top\hat{G}_2^\top & \mathcal{C}\mathcal{C}^\top + \hat{G}_1\hat{G}_1^\top \end{bmatrix},$$

$$= \begin{bmatrix} \mathcal{O}\mathcal{O}^\top + \hat{G}_2\hat{G}_2^\top & 0 \\ 0 & \mathcal{C}\mathcal{C}^\top + \hat{G}_1\hat{G}_1^\top \end{bmatrix} + \begin{bmatrix} 0 & \hat{G}_2\hat{G}_1 - \mathcal{O}\mathcal{C} \\ \hat{G}_1^\top\hat{G}_2^\top - \mathcal{C}^\top\mathcal{O}^\top & 0 \end{bmatrix}. \tag{E.4}$$

Applying Weyl's inequality on the matrix decomposition in (E.4), we have

$$\sigma_{2n}\left(\begin{bmatrix} \mathcal{O} & \hat{G}_2 \\ -\mathcal{C}^\top & \hat{G}_1^\top \end{bmatrix}\begin{bmatrix} \mathcal{O} & \hat{G}_2 \\ -\mathcal{C}^\top & \hat{G}_1^\top \end{bmatrix}^\top\right) = \sigma_{2n}\left(\begin{bmatrix} \mathcal{O}\mathcal{O}^\top + \hat{G}_2\hat{G}_2^\top & 0 \\ 0 & \mathcal{C}\mathcal{C}^\top + \hat{G}_1\hat{G}_1^\top \end{bmatrix}\right)$$

$$- \left\|\begin{bmatrix} 0 & \hat{G}_2\hat{G}_1 - \mathcal{O}\mathcal{C} \\ \hat{G}_1^\top\hat{G}_2^\top - \mathcal{C}^\top\mathcal{O}^\top & 0 \end{bmatrix}\right\|,$$

$$= \sigma_{2n}\left(\begin{bmatrix} \mathcal{O}\mathcal{O}^\top + \hat{G}_2\hat{G}_2^\top & 0 \\ 0 & \mathcal{C}\mathcal{C}^\top + \hat{G}_1\hat{G}_1^\top \end{bmatrix}\right)$$

$$- \|\hat{G}_2\hat{G}_1 - \mathcal{O}\mathcal{C}\|,$$

$$\geq \sigma_{2n}\left(\begin{bmatrix} \mathcal{O}\mathcal{O}^\top & 0 \\ 0 & \mathcal{C}\mathcal{C}^\top \end{bmatrix}\right) - \|\hat{G}_2\hat{G}_1 - \mathcal{O}\mathcal{C}\|,$$

$$\geq \frac{1}{2}\min\left\{\sigma_n\left(\mathcal{O}\mathcal{O}^\top\right), \sigma_n\left(\mathcal{C}\mathcal{C}^\top\right)\right\} \tag{E.5}$$

Applying Tu et al. (2016, Lemma 5.4) to the matrices $\begin{bmatrix} \mathcal{O} & \hat{G}_2 \\ \mathcal{C}^\top & -\hat{G}_1^\top \end{bmatrix}$, and $\begin{bmatrix} \mathcal{O} & \hat{G}_2 \\ -\mathcal{C}^\top & \hat{G}_1^\top \end{bmatrix}$, and utilizing (E.3) and (E.5), we have

$$\text{dist}^2\left(\begin{bmatrix} \mathcal{O} & \hat{G}_2 \\ \mathcal{C}^\top & -\hat{G}_1^\top \end{bmatrix}, \begin{bmatrix} \mathcal{O} & \hat{G}_2 \\ -\mathcal{C}^\top & \hat{G}_1^\top \end{bmatrix}\right) \leq \frac{4}{\sqrt{2}-1}\frac{\left\|\hat{G}_2\hat{G}_1 - \mathcal{O}\mathcal{C}\right\|_F^2}{\min\left\{\sigma_n\left(\mathcal{O}\mathcal{O}^\top\right), \sigma_n\left(\mathcal{C}\mathcal{C}^\top\right)\right\}} \tag{E.6}$$

The proof completes by following similar arguments as used by the proof of Lemma 5.14 in Tu et al. (2016), to show that,

$$\text{dist}^2\left(\begin{bmatrix} \mathcal{O} & \hat{G}_2 \\ \mathcal{C}^\top & -\hat{G}_1^\top \end{bmatrix}, \begin{bmatrix} \mathcal{O} & \hat{G}_2 \\ -\mathcal{C}^\top & \hat{G}_1^\top \end{bmatrix}\right) = 2 \cdot \text{dist}^2\left(\begin{bmatrix} \mathcal{O} \\ \mathcal{C}^\top \end{bmatrix}, \begin{bmatrix} \hat{G}_2 \\ \hat{G}_1^\top \end{bmatrix}\right) \tag{E.7}$$

□

### E.2. Finalizing the Proof of Theorem 4

Under the setting of Lemma 6, there exists a similarity transform matrix $S$ such that,

$$\left\|\mathcal{C} - S\hat{G}_1\right\|_F^2 \leq \frac{2}{\sqrt{2}-1}\frac{\left\|\hat{G}_2\hat{G}_1 - \mathcal{O}\mathcal{C}\right\|_F^2}{\min\left\{\sigma_n\left(\mathcal{O}\mathcal{O}^\top\right), \sigma_n\left(\mathcal{C}\mathcal{C}^\top\right)\right\}}, \tag{E.8}$$

This implies that, for any new sequence of observed inputs-outputs $\{(u_\tau, y_\tau)\}_{\tau=0}^{t-1}$, let $\hat{x}_t$ be the Kalman filter estimate. Then there is a similarity transform $S$ such that, with probability at least $1 - \delta$, we have

$$\left\| \hat{x}_t - S\hat{G}_1 \bar{z}_t \right\|_{\ell_2}^2 \leq \left\| \hat{x}_t - \mathcal{C}\bar{z}_t + \mathcal{C}\bar{z}_t - S\hat{G}_1 \bar{z}_t \right\|_{\ell_2}^2,$$

$$\leq 2 \underbrace{\| \hat{x}_t - \mathcal{C}\bar{z}_t \|_{\ell_2}^2}_{\text{approximation error}} + 2 \underbrace{\left\| \left(\mathcal{C} - S\hat{G}_1\right) \bar{z}_t \right\|_{\ell_2}^2}_{\text{estimation error}}, \tag{E.9}$$

Note that, under the assumptions made in the statement of Theorem 4, we have

$$\| \hat{x}_t - \mathcal{C}\bar{z}_t \|_{\ell_2}^2 = \left\| \bar{A}^L \hat{x}_{t-L} \right\|_{\ell_2}^2 \leq \| \bar{A}^L \|^2 \| \hat{x}_{t-L} \|_{\ell_2}^2,$$

$$\leq c\, C_\rho^2 \rho^{2L} \| \hat{x}_{t-L} \|_{\ell_2}^2. \tag{E.10}$$

Combining this with (E.9), we have

$$\left\| \hat{x}_t - S\hat{G}_1 \bar{z}_t \right\|_{\ell_2}^2 \leq c\, C_\rho^2 \rho^{2L} \| \hat{x}_{t-L} \|_{\ell_2}^2 + \left\| \mathcal{C} - S\hat{G}_1 \right\|_F^2 \| \bar{z}_t \|_{\ell_2}^2,$$

$$\leq c\, C_\rho^2 \rho^{2L} \| \hat{x}_{t-L} \|_{\ell_2}^2 + \frac{2}{\sqrt{2}-1} \frac{\left\| \hat{G}_2 \hat{G}_1 - \mathcal{O}\mathcal{C} \right\|_F^2}{\min \left\{ \sigma_n \left(\mathcal{O}\mathcal{O}^\top\right), \sigma_n \left(\mathcal{C}\mathcal{C}^\top\right) \right\}} \| \bar{z}_t \|_{\ell_2}^2,$$

$$\leq \frac{4}{\sqrt{2}-1} \frac{\left\| \hat{G}_2 \hat{G}_1 - \mathcal{O}\mathcal{C} \right\|_F^2}{\min \left\{ \sigma_n \left(\mathcal{O}\mathcal{O}^\top\right), \sigma_n \left(\mathcal{C}\mathcal{C}^\top\right) \right\}} \| \bar{z}_t \|_{\ell_2}^2, \tag{E.11}$$

where we obtained the last inequality by choosing $L > 0$ such that

$$c\, C_\rho^2 \rho^{2L} \| \hat{x}_{t-L} \|_{\ell_2}^2 \leq \frac{2}{\sqrt{2}-1} \frac{\left\| \hat{G}_2 \hat{G}_1 - \mathcal{O}\mathcal{C} \right\|_F^2}{\min \left\{ \sigma_n \left(\mathcal{O}\mathcal{O}^\top\right), \sigma_n \left(\mathcal{C}\mathcal{C}^\top\right) \right\}} \| \bar{z}_t \|_{\ell_2}^2,$$

$$\impliedby L \geq \beta \log \left( \frac{c\, C_\rho^2 \| \hat{x}_{t-L} \|_{\ell_2}^2 \min \left\{ \sigma_n \left(\mathcal{O}\mathcal{O}^\top\right), \sigma_n \left(\mathcal{C}\mathcal{C}^\top\right) \right\}}{\left\| \hat{G}_2 \hat{G}_1 - \mathcal{O}\mathcal{C} \right\|_F^2 \| \bar{z}_t \|_{\ell_2}^2} \right) (1 - \rho)^{-1}.$$

plugging in the upper bounds on $\left\| \hat{G}_2 \hat{G}_1 - \mathcal{O}\mathcal{C} \right\|_F^2$ and $\| \bar{z}_t \|_{\ell_2}^2$ from Theorem 3, and Lemma 8 gives us the statement of Theorem 4 and completes the proof.

**Remark 2.** *If additionally, Assumption 1 also holds for the new sequence of observed inputs-outputs $\{(u_\tau, y_\tau)\}_{\tau=0}^{t-1}$, then from (E.10), with probability at least $1 - \delta$, we have*

$$\| \hat{x}_t - \mathcal{C}\bar{z}_t \|_{\ell_2}^2 = \left\| \bar{A}^L \hat{x}_{t-L} \right\|_{\ell_2}^2 \leq c\, C_\rho^2 \rho^{2L} \| \Sigma[\hat{x}_t] \| \left( n + \log(t/\delta) \right) \tag{E.12}$$

*where we get the last inequality by applying Lemma 8. This gives us the statement of Proposition 1. Hence, for such a sequence we have,*

$$\left\| \hat{x}_t - S\hat{G}_1 \bar{z}_t \right\|_{\ell_2}^2 \leq c\, C_\rho^2 \rho^{2L} \| \Sigma[\hat{x}_t] \| \left( n + \log(t/\delta) \right) + \frac{2}{\sqrt{2}-1} \frac{\left\| \hat{G}_2 \hat{G}_1 - \mathcal{O}\mathcal{C} \right\|_F^2}{\min \left\{ \sigma_n \left(\mathcal{O}\mathcal{O}^\top\right), \sigma_n \left(\mathcal{C}\mathcal{C}^\top\right) \right\}} \| \bar{z}_t \|_{\ell_2}^2,$$

$$\leq \frac{4}{\sqrt{2}-1} \frac{\left\| \hat{G}_2 \hat{G}_1 - \mathcal{O}\mathcal{C} \right\|_F^2}{\min \left\{ \sigma_n \left(\mathcal{O}\mathcal{O}^\top\right), \sigma_n \left(\mathcal{C}\mathcal{C}^\top\right) \right\}} \| \bar{z}_t \|_{\ell_2}^2, \tag{E.13}$$

*where we obtained the last inequality by choosing $L > 0$ such that*

$$c\, C_\rho^2 \rho^{2L} \|\Sigma[\hat{x}_t]\| \left(n + \log(t/\delta)\right) \leq \frac{2}{\sqrt{2} - 1} \frac{\left\|\hat{G}_2 \hat{G}_1 - \mathcal{OC}\right\|_F^2}{\min\left\{\sigma_n\left(\mathcal{OO}^\top\right), \sigma_n\left(\mathcal{CC}^\top\right)\right\}} \|\bar{z}_t\|_{\ell_2}^2,$$

$$\Longleftarrow L \geq \beta \log \left( \frac{c\, C_\rho^2 \|\Sigma[\hat{x}_t]\| \left(n + \log(t/\delta)\right) \min\left\{\sigma_n\left(\mathcal{OO}^\top\right), \sigma_n\left(\mathcal{CC}^\top\right)\right\}}{\left\|\hat{G}_2 \hat{G}_1 - \mathcal{OC}\right\|_F^2 \|\bar{z}_t\|_{\ell_2}^2} \right) (1 - \rho)^{-1}.$$

*plugging in the upper bounds on $\left\|\hat{G}_2 \hat{G}_1 - \mathcal{OC}\right\|_F^2$ from Theorem 3, and $\|\bar{z}_t\|_{\ell_2}^2$ from Lemma 8, we obtain (a modified) statement of Theorem 4 with $\|\hat{x}_{t-L}\|_{\ell_2}^2$ replaced by $\|\Sigma[\hat{x}_t]\| \left(n + \log(t/\delta)\right)$, and $\|\bar{z}_t\|_{\ell_2}^2$ replaced by $(\|C\Sigma[\hat{x}_t]C^\top + \Sigma_e\| + \|\Sigma_u\|)\left((m + p)L + \log(2t/\delta)\right)$, when Assumption 1 also holds for the new sequence of observed inputs-outputs $\{(u_\tau, y_\tau)\}_{\tau=0}^{t-1}$.*

## F. Optimization Landscape (Proof of Proposition 2)

*Proof.* The proof of Proposition 2 relies on Theorem 3 of Zhu et al. (2020) which requires that $\sum_{t=1}^T \bar{z}_t \bar{z}_t^\top$ has full rank. Recall that, Theorem 6 ensures that choosing $L \gtrsim \beta \log\left(C_\rho T \|C\| \sqrt{C_{\bar{z}}(\delta) C_{\hat{x}}(\delta)}/c \log(T)\right)/(1 - \rho)$, and

$$T \gtrsim L\left((m + p)L\, \log\left(\frac{2T \|\Sigma[\bar{z}_T]\|}{3L\lambda_{\min}\left(\Sigma[\bar{z}_L]\right)}\right) + \log(1/\delta)\right),$$

the event $\mathcal{E} := \{\sum_{t=1}^T \bar{z}_t \bar{z}_t^\top \succ 0\}$ holds with probability at least $1 - \delta$. Thus, under the event $\mathcal{E}$, applying Theorem 3 of (Zhu et al., 2020) gives us immediately the desired statements of Proposition 2. $\square$

## G. Supporting Lemmas

In this section, we provide a list of auxiliary lemmas that will be useful to derive our main results.

**Lemma 7** (Sub-exponential tail). *Suppose $x \sim \mathcal{N}(0, \Sigma_x)$ with $\Sigma_x \in \mathbb{R}^{d_x \times d_x}$. For any $\rho \geq (3 + 2\sqrt{2})d_x$, we have*

$$\mathbb{P}(\|x\|_{\ell_2}^2 \geq 3\|\Sigma_x\|\rho) \leq e^{-\rho}.$$

*Proof.* From (Hsu et al., 2012, Proposition 1), we have for any $\rho > 0$,

$$\mathbb{P}(\|x\|_{\ell_2}^2 \geq \text{tr}(\Sigma_x) + 2\sqrt{\text{tr}(\Sigma_x^2)\rho} + 2\|\Sigma_x\|\rho) \leq e^{-\rho},$$

which implies

$$\mathbb{P}(\|x\|_{\ell_2}^2 \geq d_x\|\Sigma_x\| + 2\sqrt{d_x}\|\Sigma_x\|\sqrt{\rho} + 2\|\Sigma_x\|\rho) \leq e^{-\rho}.$$

We can see that when $\rho \geq (3 + 2\sqrt{2})d_x$, we have $d_x + 2\sqrt{d_x}\sqrt{\rho} \leq \rho$, which further implies that $d_x\|\Sigma_x\| + 2\sqrt{d_x}\|\Sigma_x\|\sqrt{\rho} \leq \|\Sigma_x\|\rho$. Therefore, we have $\mathbb{P}(\|x\|_{\ell_2}^2 \geq 3\|\Sigma_x\|\rho) \leq e^{-\rho}$. $\square$

Using Lemma 7, we can upper bound the squared Euclidean norm of $\{\hat{x}_t\}_{t=1}^T$, $\{\bar{z}_t\}_{t=1}^T$, $\{\xi_t\}_{t=1}^T$, as follows.

**Lemma 8** (Bounded States). *Fix a failure probability $\delta > 0$. Suppose Assumption 1 holds, and let $\Sigma[\hat{x}_t] := \mathbb{E}\left[\hat{x}_t \hat{x}_t^\top\right] = \sum_{k=0}^{t-1} A^k B \Sigma_u B^\top (A^\top)^k + \sum_{k=0}^{t-1} A^k F \Sigma_e F^\top (A^\top)^k$ denote the covariance of the predicted state $\hat{x}_t$ given by (3.2). There exists universal constant $c > 0$ such that,*

$$\mathbb{P}\left(\bigcap_{t=1}^T \left\{\|\hat{x}_t\|_{\ell_2}^2 \leq c\|\Sigma[\hat{x}_T]\| \left(n + \log(T/\delta)\right)\right\}\right) \geq 1 - \delta,$$

$$\mathbb{P}\left(\bigcap_{t=1}^T \left\{\|\bar{z}_t\|_{\ell_2}^2 \leq c(\|C\Sigma[\hat{x}_T]C^\top + \Sigma_e\| + \|\Sigma_u\|)\left((m + p)L + \log(2T/\delta)\right)\right\}\right) \geq 1 - \delta,$$

$$\mathbb{P}\left(\bigcap_{t=1}^T \left\{\|\xi_t\|_{\ell_2}^2 \leq c\|\mathcal{T}_u(\Sigma_u \otimes I)\mathcal{T}_u^\top + \mathcal{T}_e(\Sigma_e \otimes I)\mathcal{T}_e^\top\|\left(mH + \log(2T/\delta)\right)\right\}\right) \geq 1 - \delta. \tag{G.1}$$

*Proof.* Recall from (3.2) that,

$$\hat{x}_{t+1} = A\hat{x}_t + Bu_t + Fe_t \implies \hat{x}_t = \sum_{k=0}^{t-1} A^{t-1-k}(Bu_k + Fe_k) \tag{G.2}$$

Therefore, under Assumption 1, for all $t \geq 0$, we have $\mathbb{E}[\hat{x}_t] = 0$ and

$$\Sigma[\hat{x}_t] := \mathbb{E}\left[\hat{x}_t\hat{x}_t^\top\right] = \sum_{k=0}^{t-1} A^k B\Sigma_u B^\top (A^\top)^k + \sum_{k=0}^{t-1} A^k F\Sigma_e F^\top (A^\top)^k \tag{G.3}$$

Hence, using Lemma 7, together with union bound over all $t \in [1, T]$, and for any $\rho \geq (3 + 2\sqrt{2})n$, we have

$$\mathbb{P}\left(\bigcap_{t=1}^{T} \left\{ \|\hat{x}_t\|_{\ell_2}^2 \leq 3\|\Sigma[\hat{x}_t]\|\rho \right\}\right) \geq 1 - Te^{-\rho} \tag{G.4}$$

Choosing $\rho = (3 + 2\sqrt{2})n + \log(T/\delta)$, we have,

$$\mathbb{P}\left(\bigcap_{t=1}^{T} \left\{ \|\hat{x}_t\|_{\ell_2}^2 \leq 3\|\Sigma[\hat{x}_t]\| \left((3 + 2\sqrt{2})n + \log(T/\delta)\right) \right\}\right) \geq 1 - \delta. \tag{G.5}$$

Noting that $\Sigma[\hat{x}_{t+1}] \succeq \Sigma[\hat{x}_t]$ for all $t \geq 0$, we have

$$\mathbb{P}\left(\bigcap_{t=1}^{T} \left\{ \|\hat{x}_t\|_{\ell_2}^2 \leq 3\|\Sigma[\hat{x}_T]\| \left((3 + 2\sqrt{2})n + \log(T/\delta)\right) \right\}\right) \geq 1 - \delta. \tag{G.6}$$

Using similar argument as in (G.6), it is easy to show that,

$$\mathbb{P}\left(\bigcap_{t=1}^{T} \left\{ \|u_t\|_{\ell_2}^2 \leq 3\|\Sigma_u\| \left((3 + 2\sqrt{2})p + \log(T/\delta)\right) \right\}\right) \geq 1 - \delta, \tag{G.7}$$

$$\mathbb{P}\left(\bigcap_{t=1}^{T} \left\{ \|y_t\|_{\ell_2}^2 \leq 3\|C\Sigma[\hat{x}_T]C^\top + \Sigma_e\| \left((3 + 2\sqrt{2})m + \log(T/\delta)\right) \right\}\right) \geq 1 - \delta. \tag{G.8}$$

To upper bound the Euclidean norm of $\bar{z}_t = [y_{t-1}^\top \cdots y_{t-L}^\top u_{t-1}^\top \cdots u_{t-L}^\top]^\top$, note that, $\|\bar{z}_t\|_{\ell_2}^2 = \sum_{\tau=t-L}^{t-1} \|y_\tau\|_{\ell_2}^2 + \sum_{\tau=t-L}^{t-1} \|u_\tau\|_{\ell_2}^2$. Combining this with (G.7) and (G.8), we get

$$\mathbb{P}\left(\bigcap_{t=1}^{T} \left\{ \|\bar{z}_t\|_{\ell_2}^2 \leq 3(\|C\Sigma[\hat{x}_T]C^\top + \Sigma_e\| + \|\Sigma_u\|) \left((3 + 2\sqrt{2})(m + p)L + \log(2T/\delta)\right) \right\}\right) \geq 1 - \delta. \tag{G.9}$$

Using similar line of reasoning, for $\xi_t = \mathcal{T}_u u_{t:t+H-2} + \mathcal{T}_e e_{t:t+H-1}$, we have

$$\mathbb{P}\left(\bigcap_{t=1}^{T} \left\{ \|\xi_t\|_{\ell_2}^2 \leq 3\|\mathcal{T}_u(\Sigma_u \otimes I)\mathcal{T}_u^\top + \mathcal{T}_e(\Sigma_e \otimes I)\mathcal{T}_e^\top\| \left((3 + 2\sqrt{2})mH + \log(2T/\delta)\right) \right\}\right) \geq 1 - \delta. \tag{G.10}$$

This completes the proof. $\qquad\square$

# H. Experimental Details

In this section, we describe the hyperparameter details and Python/PyTorch pseudocode of experiments in Section 5. The code can be found in https://github.com/sdean-group/linear-ar-kf. The following describes a pseudocode of training the auto-regressive model.

```
# train model

torch.manual_seed(torch_seed)
model = TwoLayerLinearAR((m+p)*L, n, m*H)

criterion = nn.MSELoss()
optimizer = optim.Adam(model.parameters(), lr=lr, weight_decay=weight_decay)
scheduler = StepLR(optimizer, step_size=step_size, gamma=gamma)

model.train()
for epoch in range(epochs):
    for batch_inputs, batch_outputs in train_loader:
        outputs = model(batch_inputs)
        loss = criterion(outputs, batch_outputs)

        optimizer.zero_grad()
        loss.backward()
        optimizer.step()

    scheduler.step()
```

### H.1. Architecture Search on Synthetic Systems

For architecture search over the hidden dimension, we use $L = 10$, $H = 5$, $T_{\text{train}} = 10^4$, $T_{\text{test}} = 10^3$, `epochs`= 150, `batch_size`= 64, `lr`= 0.01, `step_size`= 1, `gamma`= 0.9, and `weight_decay`= $10^{-3}$.

### H.2. Latent State Recovery

For training the two-layer neural network to generate the scatter plot in Figure 2, we use $L = 10$, $H = 5$, $T_{\text{train}} = 10^4$, $T_{\text{test}} = 10^3$, `epochs`= 150, `batch_size`= 64, `lr`= 0.05, `step_size`= 2, `gamma`= 0.9, and `weight_decay`= $10^{-3}$. Figure 2 is generated with the pairs (`x_kf`, `transformed_activated_states`) where the variables are defined in the pseudocode below.

```
# Check Alignment

G1 = model.linear1.weight.detach().numpy()
activated_states = inputs_from_test_trajectory @ G1.T
x_kf = KF_predicted_state_estimates_from_test_trajectory

coeff, _, _, _ = np.linalg.lstsq(activated_states, X_kf, rcond=None)
transformed_activated_states = activated_states @ coeff
```

We also compute $R^2$ values coordinate-wise in Table 2. For each coordinate $i \in \{1, \ldots, n\}$, $R^2$ value is computed as

$$R_i^2 = 1 - \frac{\sum_{t=L}^{T_{\text{test}}} \left( \hat{x}_{t,i} - \hat{S}\hat{G}_1 \bar{z}_{t,i} \right)^2}{\sum_{t=L}^{T_{\text{test}}} \left( \hat{x}_{t,i} - \bar{x}_i \right)^2}, \quad \text{where} \quad \bar{x}_i = \frac{1}{T_{\text{test}} - L + 1} \sum_{t=L}^{T_{\text{test}}} \hat{x}_{t,i}.$$

The mean $R^2$ is computed as $\frac{1}{n} \sum_{i=1}^{n} R_i^2$.

### H.3. ControlGym Experiments

For the hyperparameters in this experiment, they are the same as in the synthetic experiments. We use two benchmark environments from ControlGym (Zhang et al., 2024): the underwater vehicle environment (ID: `umv`) and the aircraft environment (ID: `ac6`). Noise processes $\{w_t\}_{t \geq 0}$ and $\{v_t\}_{t \geq 0}$ are also defined in the environment. We note that, for the underwater vehicle environment (ID: `umv`), the measurement-noise covariance is zero. For the aircraft environment (ID: `ac6`), both process noise and measurement noise are nonzero and independent of each other. In this experiment, the covariance is rescaled by multiplying 0.05.

