# OpenReview forum: "Two-Layer Linear Auto-Regressive Models Estimate Latent States"
_ICML.cc/2026/Conference — ICML 2026 regular_

### Official Review · Reviewer_tePB · 2026-03-11

**Soundness:** 3
**Presentation:** 4
**Significance:** 2
**Originality:** 2
**Overall Recommendation:** 4
**Confidence:** 2

**Summary:**

This paper presents a result that characterises when a linear autoregressive model approximates a the Kalmar filter updates. The paper shows that the hidden latent representation from a Empirical Risk Minimisation perspective corresponds to the Kalman filter. The paper further shows that the non-convex learning problem is somewhat well behaved and show experimentally that the learned updates are very closely correlated with the Kalman updates. The paper is well written and easy to follow, while some of the mathematical concepts are quite dense the authors make a good job of providing a narrative for the reader that is accessible on several abstraction levels.

**Compliance With Llm Reviewing Policy:**

Affirmed.

**Key Questions For Authors:**

- The convergence guarantees in 4.2 are as you say for an unconstrained problem, while showing this for the specific constraints in this set-up feels challenging could you provide more intuition of how relevant you believe this result actually is for your problem setting?

**Limitations:**

yes

**Strengths And Weaknesses:**

- Strengths
  - This is a well written paper that to my knowledge attributes the relevant related work and nicely contributes to a body of work and advances on the state of the art. The introduction really sets up and motivates the problem well and the theoretical justification of the work is good.
  - While the paper is theoretical in nature the empirical results are clear and provides intuition that covers the "leaps of faith" that have been necessary.
- Weaknesses
  - I would recommend moving the actual learning problem that is written under Eq. 3.5 to the experimental section as it disturbs the flow of the paper at this point. The main results are related to the assumption that we have the global minimiser of 3.5 and the statements about the learning problem with an inner and outer loop can be left to the experimental section. That will create a better flow of the paper and allow us to first think about the ideal scenario before moving on to what is practically possible coming in 4.2.
  - While I find the results interesting and it fits into a narrative and a body of previous work. One of the weaknesses of this work is that it is probably of quite limited interest to a large part of the community.

---

> ### Author Rebuttal · Authors · 2026-03-31
>
> Thank you for your valuable review of our paper!
>
> > **Q1:** Convergence guarantees
>
> Thank you for raising this point! We agree that convergence guarantees for constrained non-convex optimization would be highly valuable. Establishing convergence for gradient-based methods in this setting appears to be technically challenging in general. Prior work has shown, for example, that projected gradient descent may converge to saddle points, and that certifying second-order stationarity for constrained non-convex problems is computationally hard [8]. In view of these difficulties, we believe that obtaining convergence guarantees likely requires additional structural assumptions on the loss landscape and/or the constraint set. Investigating whether suitably perturbed projected gradient-type methods can yield such guarantees is an interesting direction for future work, but it is beyond the scope of the present paper.
>
> > **Q2:** Moving actual learning problem
>
> Thanks for the suggestion! We have moved the discussion of how we (practically) solve the ERM problem (3.5) to Section 4.2 (Optimization results). Note that only the first two sentences after Eqn (3.5) discuss how we (practically) solve the non-convex ERM problem. We believe that it is important to keep the discussion after the first two sentences (starting from line 167), since it helps in understanding the ERM problem (3.5). Specifically, we discuss the purpose of norm bound, and draw a connection with training a two layer linear network with weight decay. Delaying this discussion to the experimental section would make it less coherent.
>
> > **Q3:** Limited interest
>
> We respectfully disagree with the reviewer’s claim. As pointed out by the **Reviewer xuxP**, “theoretical analysis of autoregressive models is timely and important – even for a restricted class of 2-layer linear architectures – as these form the backbone of many modern sequence models.”
>
> More broadly, we believe the conceptual relevance of our work extends well beyond Kalman filtering itself. The central question is not merely whether an autoregressive model can emulate a Kalman filter, but whether end-to-end autoregressive training can recover meaningful latent-state representations from partially observed sequential data (specifically, if the sequential data has some hidden structure). This problem is of significant interest to the machine learning community, and has been explored empirically for autoregressive output prediction of hidden Markov models (HMMs) using Transformers [10]. Whether the Transformers internally learn (at the softmax activations)  some belief states corresponding to the latent states of HMMs is still an open problem both empirically as well as theoretically. The linear dynamical system setting provides a rare regime where this question can be answered rigorously.
>
> Furthermore, Kalman filtering itself is one of the canonical frameworks for latent-state inference from sequential data, and state-space methods continue to play an important role across robotics, tracking, signal processing, time-series analysis, and econometrics. More recently, related state-space ideas have also influenced modern sequence-modeling architectures, including models such as Mamba [11]. In this sense, we view our work not as addressing a narrow special case, but as offering a principled theoretical lens on representation learning in autoregressive sequence models.
>
> # References:
> [1] Komaroff, Iterative matrix bounds and computational solutions to the discrete algebraic riccati equation. IEEE TAC 2002.
>
> [2] Lale et al. Logarithmic regret bound in partially observable linear dynamical systems. NeurIPS 2020.
>
> [3] Talebi et al. Data-driven optimal filtering for linear systems with unknown noise covariances. NeurIPS 2023.
>
> [4] Tsiamis et al. Online learning of the kalman filter with logarithmic regret. IEEE TAC 2022.
>
> [5] Ghai et al. No-regret prediction in marginally stable systems. COLT 2020.
>
> [6] Bertsekas, Dynamic programming and optimal control. Athena scientific, 2012.
>
> [7] Kumar et al. Stochastic systems: Estimation, identification, and adaptive control. SIAM, 2015.
>
> [8] Nouiehed et al. Convergence to second-order stationarity for constrained non-convex optimization. arXiv:1810.02024.
>
> [9] Kawaguchi. Deep learning without poor local minima. NeurIPS 2016.
>
> [10] Dai et al. Pre-trained large language models learn to predict hidden markov models in-context. NeurIPS 2025.
>
> [11] Gu et al. Mamba: Linear-time sequence modeling with selective state spaces. COLM 2024.
>
> [12] Tadipatri et al. Nonconvex linear system identification with minimal state representation. L4DC 2025.
>
> [13] Haeffele et al. Global optimality in neural network training. CVPR 2017.
>
> [14] Sarkar et al. Finite Time LTI System Identification. JMLR 2021.
>
> [15] Tsiamis et al. Sample complexity of kalman filtering for unknown systems. L4DC 2020.
>
> [16] Mania et al. Time varying regression with hidden linear dynamics. L4DC 2022.

---

> > ### Author Rebuttal · Reviewer_tePB · 2026-04-07
> >
> > Thank you for the comments, I will retain my score.

---

> > > ### Author Response · Authors · 2026-04-08
> > >
> > > We sincerely thank the reviewer for the thoughtful review and for taking the time to read our rebuttal carefully!
> > >
> > > We appreciate the reviewer’s suggestion to move the discussion of how we (practically) solve the ERM problem (3.5) to Section 4.2 (Optimization results). We also appreciate the questions asked by the reviewer regarding the convergence guarantee and the significance of our work.
> > >
> > > We will incorporate our response to the reviewer’s questions into the revision, and we believe it will significantly improve the clarity of the paper.

---

### Official Review · Reviewer_vqtk · 2026-03-13

**Soundness:** 4
**Presentation:** 4
**Significance:** 3
**Originality:** 3
**Overall Recommendation:** 5
**Confidence:** 3

**Summary:**

The paper presents a theoretical analysis of a linear neural network for forecasting future states of a linear dynamical system. It is shown that the latent representation of the network coincides, up to a similarity transform, with the states estimated by the Kalman filter. The authors also present a result on the optimization landscape of the problem, stating that any local minima are actually the global minimum with high probability.

**Compliance With Llm Reviewing Policy:**

Affirmed.

**Final Justification:**

The information provided in the authors' response would further strengthen the paper. I keep my originally positive score (5).

**Key Questions For Authors:**

In Theorems 1 & 3 and Proposition 2 (and perhaps somewhere else too), the bounds of $T$ have $T$ in the right-hand side; for example:
$$T \gtrsim L(d_{\bar{z}} + \log(T / \delta))$$
in Theorem 1. Is this representation inevitable?

**Limitations:**

Yes.

**Strengths And Weaknesses:**

This is an interesting and enlightening paper that establishes a rigorous connection between the Kalman filter and a neural network-based prediction model. Although I did not follow all the mathematical details, the analysis sounds valid and intuitively makes a lot of sense. The paper is so well written that I could follow most of the reasoning although I am not a very expert in this kind of theoretical analysis. The paper presents an important piece of result toward understanding the internal behavior of more general, larger neural networks.

I found no major concerns. There are a few points where the authors' intention to write so is unclear, mostly due to some typos or violated notations. Below are small points that I don't need particular responses:
- In Theorem 1 and Theorem 4, the test data, {$(u_\tau, y_\tau)$}${}_{\tau=0}^{t-1}$, is defined with $t-1$ as the data size, and then $t$ is used as the timestamp of the prediction target. It is a little confusing because small $t$ is used elsewhere to denote a timestamp generally.
- What is $\rho$ without argument that appears in the main theorem?
- Page 4, Lines 171-172 (left): "In deep learning practice, **is it** common to implement ..." --> it is

Reading the paper a thought came up, with which I don't intend to question the value of the paper. To me the main result, i.e., the linear AR network's latent coincides with the Kalman filter estimation, is somewhat intuitive and not very surprising, because Kalman filter is optimal for linear Gaussian systems, and the network is now learned indeed on data from a linear Gaussian system by optimizing the prediction error. So I am wondering how the result should be positioned; is it like the equivalence is something surprising, or is it somewhat anticipated naturally and the paper's contribution rather lies in the rigorous theoretical analysis showing the connection? A bit more explanation on this aspect might make the paper's positioning clearer.

---

> ### Author Rebuttal · Authors · 2026-03-31
>
> Thank you for your positive remarks and valuable review of our paper! We have address each of your points below.
>
> > **Q1:**  timestamp of the prediction target
>
> Thanks for the suggestion! To avoid confusion, we now use the time index $t'$, and the trajectory length $T_{test}$ for the test data in Theorems 1 and 4.
>
> > **Q2:**  $\rho$ without argument
>
> Thanks for asking this question! The $\rho$ without argument is a fixed scalar strictly between the spectral radius $\rho(\bar A)$ and $1$. It is the geometric decay rate used to bound powers of closed-loop estimator matrix $\bar A$. In our paper, $\rho$ is defined right before Proposition 1 in Section 4.1. To avoid confusion, we have moved the definition of $\rho$ and placed it before Theorem 1 (after line 202), and have replaced $\rho$ with $\tilde{\rho}$.
>
> > **Q3:**  Typo. "is it"
>
> Thanks for pointing this out. Fixed!
>
> > **Q4:**   paper's positioning
>
> Great question! Please note that while the emergence of a Kalman-like predictor may be heuristically expected in the linear-Gaussian setting, it is not a priori clear that end-to-end prediction training should recover the **internal latent representation** of the Kalman filter, despite having no access to the parameters $A, B, C, \Sigma_w, \Sigma_v$. It is also not clear a priori that this internal latent representation learning of Kalman filter occurs at global (or local) optima of a non-convex objective with finite samples.
>
> Hence, the value of our work is in proving that **end-to-end AR training implicitly performs Kalman-state representation learning, and in characterizing when and why this happens**. This is distinct from classical system-identification pipelines that first fit a predictor and then explicitly extract latent states via Ho-Kalman-style factorization [2, 15].  In our case, the latent states estimate arises directly in the network activations. We will revise the introduction to make this positioning clearer and to emphasize that the novelty of our paper lies in the rigorous representation-learning characterization, rather than in claiming that the Kalman filtering connection is wholly unexpected in our setting.
>
> > **Q5:**  Restricted to linear Gaussian
>
> The Gaussian noise assumption is indeed useful and allows us to argue that the innovation process obtained from the Kalman filter is a sequence of independent random variables. More precisely, the proof of Theorem 2 involves bounding the Martingale offset complexity (Appendix B.2) which captures the complexity of the ERM problem (3.5). This requires the innovation error $e_t$ to be a Martingale difference sequence (much weaker condition than the Gaussian assumption). We found that, when the process and measurement noises $w_t, v_t$ are not Gaussian, the Kalman filtering innovation error $e_t$ is not a Martingale difference sequence, hence solving the ERM problem (3.5) does not give consistent estimate of the Hankel matrix $\mathcal{O}\mathcal{C}$.
>
> Prior work on sequence prediction for dynamical systems have also highlighted this as a challenging technical question (see appendix H in [5]), and treat the problem as having adversarial noise which leads to slower rates (see their Theorem 3). As it stands, generalizing our results to non-Gaussian case remains an open problem.
>
> > **Q6:**  $T \gtrsim L(d_z + log(T/ \delta))$ representation inevitable
>
> Thanks for asking this! In the statistical learning theory literature, It is very common to have such a lower bound on the minimum number of samples required [2, 14, 15, 16, 17].  The effect of $T$ appearing inside logarithm on the left hand side is extremely mild, specifically  $\log(T) << T$ for larger $T$. The $T$ appearing in our lower bound with the probability of failure $\delta$ is introduced by the union bounding the probability of failure over $T$ samples.
>
> To better understand why $T$ appears in our lower bound, we recommend reading our proof in Appendix G, where we are upper bounding the Euclidean norm of various random processes with high probability. Specifically, it is clear that the $\log(T/\delta)$ appears from applying Lemma 8 along with union bound over all samples.
>
> # References:
>
> [2] Lale et al. Logarithmic regret bound in partially observable linear dynamical systems. NeurIPS 2020.
>
> [5] Ghai et al. No-regret prediction in marginally stable systems. COLT 2020.
>
> [14] Sarkar et al. Finite Time LTI System Identification. JMLR 2021.
>
> [15] Tsiamis et al. Sample complexity of kalman filtering for unknown systems. L4DC 2020.
>
> [16] Mania et al. Time varying regression with hidden linear dynamics. L4DC 2022.
>
> [17] Krahmer et al. Suprema of chaos processes and the restricted isometry property. Communications on Pure and Applied Mathematics 2014.

---

> > ### Author Rebuttal · Reviewer_vqtk · 2026-04-01
> >
> > Thank you for the rebuttal! I would maintain my originally positive score.

---

> > > ### Author Response · Authors · 2026-04-07
> > >
> > > We sincerely thank the reviewer for acknowledging that our initial rebuttal has fully resolved the concerns raised by the reviewer!
> > >
> > > We appreciate the questions asked by the reviewer regarding the paper's positioning, and the implications of going beyond Gaussian noise. We will fix the typos pointed out by the reviewer, and will incorporate our response to the reviewer's questions into the revision!

---

### Official Review · Reviewer_xuxP · 2026-03-17

**Soundness:** 2
**Presentation:** 3
**Significance:** 1
**Originality:** 3
**Overall Recommendation:** 3
**Confidence:** 3

**Summary:**

The paper shows that under **full observability** (rank(𝒪) = n) and **stabilizability**, the steady-state Kalman filter, which reduces to a fixed  **linear time-invariant (LTI) recurrence** under these conditions can be exactly represented
by a 2-layer linear autoregressive model. The learned hidden states are shown to coincide with  the posterior mean of the steady-state Kalman filter, with theoretical guarantees on prediction error and sample complexity. Experiments are conducted on a self created synthetic benchmark, constructed from random Gaussian matrices.

**Compliance With Llm Reviewing Policy:**

Affirmed.

**Key Questions For Authors:**

1. Please clarify the rationale behind the synthetic dataset design choices - specifically how the rescaling to ρ(A)=1 was performed  and why these particular dimensions (n=4, m=3) and noise levels were chosen.

2. Experiments on real dynamical systems from domains where Kalman filtering is widely used, robotics, neuroscience, or navigation, would significantly strengthen the empirical case and clarify whether the results hold beyond the narrow synthetic regime.

3. Do the results extend to non-steady-state conditions, where the Kalman predict-update recursion involves a time-varying gain Fₜ and is no longer a fixed linear recurrence? This is arguably the more practically relevant regime and the paper would benefit from at least a discussion of this case.

**Limitations:**

The authors are upfront about a few limitations but not all.

**Strengths And Weaknesses:**

## Strengths

1. Theoretical analysis of autoregressive models is timely and important - even for a restricted class of 2-layer linear architectures - as these form the backbone of many modern sequence models.
2. The theoretical framework and main-text derivations appear technically sound for the specific case (steady state fully observable kalman filter) studied. I have not verified the proofs in Appendix in detail.

## Weaknesses

1. Restricted theoretical scope - The result applies only when the Kalman filter has settled to steady-state with constant gain F_∞, reducing the recursion to a fixed LTI filter. Real-world Kalman filters - and those used alongside modern SSMs, rarely operate under this fixed-gain condition where Kalman recurrence is non-linear. Partial observability, transient phases, and structured noise are not addressed.

2. The core result may be circular - At steady state the Kalman recursion is already a linear RNN with fixed weights. Showing a 2-layer linear model approximates another linear recurrence is not obviously non-trivial, the two are structurally identical up to reparameterization.

3. Posterior covariance not addressed - Only the posterior mean μ_{t|t} is covered. Σ_{t|t} is trivial under the paper's assumptions but becomes the interesting quantity precisely in the excluded regimes, partial observability and uncertainty quantification.

4. Weak empirical evidence - The empirical evidence is weak and the rationale behind choosing the particular dataset and the data-generating process is not explained clearly. Why is it interesting ?

5. Unclear takeaway - The result does not characterize when the approximation breaks down, which architectural properties matter, or how insights extend beyond the studied regime. The practical or architectural implications for future work are not evident.

---

> ### Author Rebuttal · Authors · 2026-03-31
>
> Thank you for your valuable review of our paper!
>
> > **Q1:**  Restricted theoretical scope
>
> Regarding **steady state Kalman filtering and transient phases**, we refer the reviewer to our response to **Q1 of Reviewer WBH7**.
>
> Regarding **partial observability**, we want to clarify that the impression that our work does not consider partial observability is mistaken. We indeed work with partially observed dynamical systems and do not assume invertibility of the observation matrix $C$. The condition required in Theorem 4 regarding the extended observability matrix to be full column rank does not mean that the dynamical system in Eqn (3.1) is fully observable. This assumption is standard in the system identification and Kalman filtering literature [2,5,15] to make sure the Kalman filter error covariances are uniformly bounded [7, Chapter 7]. Lastly, our results in Theorems 2 – 3 hold without this assumption.
>
> > **Q2:**  The core result may be circular
>
> We respectfully disagree with the reviewer. Training linear RNN is structurally very different from the ERM problem (3.5) we studied. The two problems differ significantly in terms of the memory requirement, computation, and loss landscape.  Moreover, if we do not exploit the structure in our problem, and treat it as a re-parametrization of a linear RNN, we will get bad dependence on the complexity of the problem. Lastly, our problem has a very different objective as compared to RNN (as we are more focused on extracting internal latent representations from the data).
>
> That being said, it is also worth noticing that our analysis can be easily extended beyond the steady-state Kalman filter. Please refer to our response to **Q1 of Reviewer WBH7**.
>
> > **Q3:**  Posterior covariance not addressed
>
> This is an interesting future direction! We agree that in the current setting, the steady state error covariance matrix can be approximated up to a similarity transform by solving a Riccati equation with the estimated matrices.
>
> The more interesting setting of directly learning error covariance from the input-output data without access to the dynamic matrices as well as noise covariances requires solving a different ERM problem [3] with different architecture, and is therefore left as a future work. One possible option is to train a model to explicitly learn the covariance matrix. This can possibly be done by regressing the future $y_t y_t^\top$ to a history of past inputs and outputs (aka the covariance method [16]).
>
> > **Q4:**  Weak empirical evidence
>
> Thanks for asking this! Please note that the major contribution of our paper is to rigorously show that end-to-end training of a two layer linear AR model can recover the hidden structure in the data generating process, with more emphasis on proving theoretical guarantees. The current empirical evidence corroborates our theory. However, we agree with the reviewer that the empirical results will be much more interesting if it involves real dynamical systems from domains where Kalman filtering is widely used. To address this useful suggestion, we repeated our experiments with ‘ControlGym’, a python library containing 36 industrial control settings. Instead of presenting the scatter plot, we present the $R^2$ statistic between the regressed learned activations and the Kalman filter predicted state estimate.
>
> # Underwater Vehicle (state dimension=8)
> | |s1|s2|s3|s4|s5|s6|s7|s8|mean $R^2$|
> |-----|-----|-----|-----|-----|-----|-----|-----|-----|-----|
> |$H=1$|0.999|0.980|0.999|0.999|0.980|0.999|1.000|0.999|0.995|
> |$H=5$|0.999|0.974|0.999|0.999|0.974|0.999|1.000|0.999|0.993|
>
> # Aircraft (state dimension=10)
> | |s1|s2|s3|s4|s5|s6|s7|s8|s9|s10|mean $R^2$|
> |-----|-----|-----|-----|-----|-----|-----|-----|-----|-----|-----|-----|
> |$H=1$|0.999|0.992|0.995|0.991|0.997|0.996|0.998|0.998|0.998|0.821|0.978|
> |$H=5$|0.999|0.991|0.996|0.991|0.996|0.995|0.997|0.998|0.997|0.836|0.979|
>
> > **Q5:**  Unclear takeaway
>
> Thanks for this comment! Please note that, our theoretical results are stated rigorously, with clearly stated conditions required for our bounds to hold. Our theoretical results do not hold when these conditions (e.g., Gaussian noise, two-layer linear network etc.) are not satisfied.  However, we agree that the practical or architectural implications for future work can be stated more clearly, and we will revise the discussion in Section 5 accordingly. We have addressed the implications of going beyond steady state Kalman filtering, and using deeper networks in our responses to **Q1 and Q2 of Reviewer WBH7**. Going beyond Gaussian noise is discussed in our response to **Q5 of Reviwer vqtk**.
>
> > **Q6:**  Design choices
>
> The spectral radius is chosen to be 1 so that the system is marginally stable, resulting in linear growth of states. This variation in state magnitude makes the state tracking ability of the hidden layer more obvious (since both vary across a wider range).
>
> # References
>
> see response to **Reviewer tePB**

---

> > ### Author Rebuttal · Reviewer_xuxP · 2026-04-04
> >
> > Thank you for the rebuttal. I appreciate the additional ControlGym experiments, which strengthen the empirical case within the studied regime. However, I maintain my overall assessment, and want to elaborate on why the steady-state assumption is more limiting than the authors suggest.
> >
> > **Time-varying Kalman gains have ubiquitous applications and are essential even in linear systems.** The authors claim that steady-state conditions hold in "most applications," but many important real-world KF deployments require time-varying gains even within the linear regime: multi-sensor fusion (sensors updating at different rates), multi-target tracking (different targets detected per scan), and intermittent observations (random sensor dropouts). Sinopoli et al. (2004, IEEE TAC) study this last setting extensively and explicitly show that the time-varying KF is optimal and tolerates higher dropout rates than its steady-state counterpart. These are all linear systems, yet steady-state is not effective because observation matrix C_t varies. Beyond these, safety-critical applications such as SLAM and spacecraft navigation also require the general KF predict-update cycle with time-varying gains.
> >
> > **General KF updates are nonlinear, not linear.** The authors' claim that extending to the general KF is straightforward deserves scrutiny. The Riccati equation for the covariance (which feeds into the mean update) is a nonlinear (fractional linear / Möbius) transformation in the precision parameters - the Kalman gain F_t depends on posterior precision through a matrix inverse, creating a rational dependence on the state (see e.g. Shaj et al. 2026, arXiv:2602.10743). A 2-layer linear autoregressive model, regardless of history length, computes a fixed linear combination of past observations and cannot implement data-dependent gain adaptation. This is an expressivity limitation, not an optimisation one - no amount of data or training will allow a fixed linear map to represent a rational transformation that can generalize. This architectural separation is further supported by Aggarwal et al. (2025, arXiv:2512.22471), who show that for Bayesian state estimation in HMMs (continuous counterparts of KF), capacity-matched MLPs trained with the same ERM objective fail by orders of magnitude while transformers recover the exact posterior. **The extension to the general KF may therefore not be possible within the authors' architectural class at all**.
> >
> > **The ERM formulation does not resolve the circularity concern.** The authors argue their formulation is structurally different from linear RNN training - but the differences work against them. A linear RNN processes arbitrarily long sequences with fixed-size weight matrices, while their $G_2G_1$ map grows with window size L. On expressivity, modern linear RNNs like Mamba or GDN support input-dependent gating, which is precisely the mechanism Aggarwal et al. (2025) show is necessary for Bayesian state estimation. The author's fixed linear map cannot implement this.
> >
> > **Recommendation.** I acknowledge the theoretical results within their scope. In my opinion, the paper's claims including the empirical setup, require further scrutiny and strengthening before acceptance. I also note that the authors are not upfront about their limitations - transparency about scope and limitations is generally encouraged and rewarded at ML conferences. I would suggest either (a) explicitly scoping the title, abstract and claims to "steady-state Kalman filters" to accurately reflect the contribution, in case of acceptance or (b) providing the claimed extension to the non-steady-state case. In general, I maintain my current recommendation - I remain unconvinced that the results, restricted to the steady-state case, have meaningful impact.
> >
> > References:
> > - Sinopoli et al. (2004). Kalman Filtering with Intermittent Observations. IEEE TAC.
> > - Shaj et al. (2026). Kalman Linear Attention. arXiv:2602.10743.
> > - Aggarwal et al. (2025). The Bayesian Geometry of Transformer Attention. arXiv:2512.22471.

---

> > > ### Author Response · Authors · 2026-04-07
> > >
> > > We sincerely thank the reviewer for the valuable feedbacks and acknowledging that the additional ControlGym experiments strengthen empirical evidence! We will add these experiments along-with our response to the reviewer's questions in the revision.
> > >
> > > > **Q7:** time-varying Kalman gains \& extension to the general KF
> > >
> > > We thank the reviewer for this clarification! We agree with the reviewer’s point that many important real-world KF deployments require time-varying gains.
> > >
> > > We want to clarify that **we did not claim that steady-state Kalman filtering holds in “most applications.”**  Instead, we argue that it is natural in our setting, which is specifically **linear time-invariant (LTI)** dynamics. For LTI systems, the KF can be time varying after initialization, but it converges exponentially to the steady state. So, the reviewers' concerns about a steady state KF restriction boils down to concerns that the LTI setting is simplistic. However, we want to emphasize that **a precise understanding of learning/filtering in LTI systems is necessary (often foundational) for developing learning/filtering performance guarantees for more general linear time-varying (LTV) dynamical systems**, as evident from the literature [18 - 21].
> > >
> > > When the dynamic matrices are **unknown as well as time-varying**, then to derive learning/filtering guarantees one requires additional assumptions (e.g. uniform stability, bounded disturbance) and/or access to multiple i.i.d. input-output trajectories of the LTV system. One can naively train a time-dependent AR model to predict H future outputs from L past inputs and output as follows: Consider the predictor form for LTV dynamics
> > >
> > > $$
> > > \hat x_{t+1} =  \bar A_t\hat x_t + B_t u_t + F_t y_t,~~~ y_t = C_t \hat x_t +  e_t.~~~ (A_t := A_t - F_t C_t)
> > > $$
> > >
> > > Define the time-varying extended observability matrix
> > >
> > > $$ \mathcal O_t :=
> > > \begin{bmatrix} C_t \\\\ C_{t+1} A_t \\\\ C_{t+2} A_{t+1} A_{t} \\\\ \vdots \\\\ C_{t+H-1} \prod_{k=t}^{t+H-2}A_k \end{bmatrix} \in \mathbb R^{mH \times n}
> > > $$
> > >
> > > Define the time-varying extended controllability matrix
> > >
> > > $$
> > > \mathcal C_t :=
> > > \begin{bmatrix}
> > >     F_{t-1} & \bar A_{t-1} F_{t-2} & \bar A_{t-1} \bar A_{t-2} F_{t-3} & \cdots &\prod_{k=t-1}^{t-L+1}A_k F_{t-L} & B_{t-1} & \bar A_{t-1} B_{t-2} & \bar A_{t-1} \bar A_{t-2} B_{t-3} & \cdots &\prod_{k=t-1}^{t-L+1}A_k B_{t-L} \end{bmatrix} \in \mathbb R^{n \times (m+p)L}
> > > $$
> > >
> > > Then, we can easily justify the approximation $\hat x_t \approx \mathcal C_t \bar z_t$ and $y_{t:t+H-1} \approx \mathcal O_t \mathcal C_t \bar z_t$, revealing an underlying structure similar to that posited by our AR model (at each time $t$).  The more efficient ways to learn the optimal LTV filtering might require alternate approaches such as **online learning, in-context learning** etc. (as referenced by the reviewer!) and is beyond the scope of our current paper.
> > >
> > > That being said, we respectfully want to emphasize that the main contribution of our paper is not to propose the best architecture for solving the optimal filtering problem. Instead, the value of our work is in proving that end-to-end AR training implicitly performs Kalman-state representation learning, despite unknown state-space matrices and noise variances, and in characterizing when and why this happens. For more detail on the positioning of our paper, please see our response to **Q4 of Reviwer vqkt**.
> > >
> > > Lastly, to avoid confusion, we will change the title of our paper to, **“Two-Layer Linear Auto-Regressive Models Estimate Latent States of LTI Systems”**
> > >
> > > > **Q8:** Circularity concern
> > >
> > > Thanks for the comments! We want to clarify that we do not claim the superiority of the two-layer AR model over RNN, and we agree with the reviewer on the superior expressivity of modern AR models like Mamba or GDN. One of the advantage of the two-layer AR model is that one can precisely characterize the optimization, estimation, and latent-representation recovery guarantees. Its main limitation is architectural rigidity. RNN have the opposite tradeoff as they are more natural for general sequence prediction, but substantially harder to analyze rigorously for optimal (Kalman) filtering.
> > >
> > > > **Q9:** Limitations
> > >
> > > The last two paragraph (indirectly) discuss the limitations of current analysis by proposing several open questions such as investigating the latent state representation learning with recurrent architectures, over-parametrized deeper networks, and transformers (in-context learning). To further clarify the limitations of our analysis, We will add a paragraph in Section 4 of the revised manuscript.
> > >
> > > # References:
> > > [18] Gradu et al. Adaptive regret for control of time-varying dynamics. L4DC 2023.
> > >
> > > [19] Minasyan et al. Online control of unknown time-varying dynamical systems. NeurIPS 202.
> > >
> > > [20] Luo et al. Dynamic regret minimization for control of non-stationary linear dynamical systems. ACM 2022.
> > >
> > > [21] Sarkar et al. Nonparametric system identification of stochastic switched linear systems.IEEE CDC 2019.

---

### Official Review · Reviewer_sRoF · 2026-03-19

**Soundness:** 2
**Presentation:** 2
**Significance:** 2
**Originality:** 2
**Overall Recommendation:** 4
**Confidence:** 2

**Summary:**

The paper aims at showing that a two layer auto-regressive model can estimate the latent state in a partially observed linear dynamical system. More precisely, the main theorem states that the auto-regressive model can recover the Kalman filter, which is the best linear filter for predicting this latent state, up to a multiplication with an invertible matrix with a large probability.
The training of the model is done by transforming the input data to an AR representation, and then minimising a square loss with respect to two matrices, whose dimensions depend on the input data and the dimension of the latent dimension of the network. It is also optimised with respect to this latent dimension.
The main result is stated informally in the main part of the paper. It is followed by some intermediate results and some experimental results. All proofs are deferred to the Appendix.

**Compliance With Llm Reviewing Policy:**

Affirmed.

**Final Justification:**

I am satisfied with the authors’ responses, and the proposed revisions will improve the paper. I changed my recommandation accordingly.

**Key Questions For Authors:**

The manuscript could benefit from a careful proofreading. Here are some examples:
- in the Related Work section, "The last of these is most closely related to our approach, though it requires stronger (Gaussian) assumptions on the noise processes" but Assumption 1 suppose the noise process is Gaussian
- the numbering in the Appendix is weird. For example, after the title "In sample Error prediction (Theorem 2)" a theorem is given and it seems very close to Theorem 2. What is the difference ?
- I am under the impression that, in several theorem, the result holds when the true dimension n of the latent space is known, e.g. in Theorem
- throughout the paper the singular values are given for matrices of the type MM^T, why don't you unify with the eigenvalues ?
- in Theorem 4, how the robustess condition is related to Theorem 3 ?
- lines 602 and 608 it seems that the same equation is displayed

**Limitations:**

In its present form, the paper lacks clarity, this makes difficult to fully assess its impact.

**Strengths And Weaknesses:**

The paper is difficult to follow in its current form due to issues with clarity and presentation. For the strength of the main result, I find it rather difficult to assess, in the present form of the paper. I think that several "constant" are hiding dependences on key parameters, e.g. $C_{\bar z}$ in Theorem 2.

Moreover I am not very convinced by the experiments. The first experiment is supposed to illustrate that the latent state is recovered but it is done with the true size of the unknown latent state and after adjustment. For the experiment on architecture search, I am a bit surprised with the results in Table 1. I do not understand how the error on the train can be smaller on the true latent dimension than on greater dimension, could the author comment more on this ?

---

> ### Author Rebuttal · Authors · 2026-03-31
>
> Thank you for your valuable review of our paper!
>
> > **Q1:** Related work
>
> Thanks for pointing this out, and we will clarify this point when revising our manuscript! Note that Lee, (2020) directly works with the innovation form and assumes that the innovation error is Gaussian white noise. We, on the other hand, assume Gaussian process, and measurement noises $w_t, v_t$. Although, this combined with Assumption 2 leads to $e_t \sim \mathcal{N}(0, C \Sigma C^\top + \Sigma_v)$ in the steady state, as discussed in our response to **Q1 of Reviewer WBH7**, when the Kalman filter is not initialized in its steady state, the innovation error will have time-varying covariance.
>
> We want to clarify that, our paper differs from Lee (2020) not in terms of noise assumptions, but in terms of the objectives: Lee (2020) is doing closed-loop system identification to recover a reduced model and bound its predictive performance relative to the finite-horizon Kalman filter. Our paper, on the other hand, is more focused on the representation learning capability of a two-layer linear auto-regressive network, trained by empirical risk minimization on data from partially observed linear dynamical systems.
>
> > **Q2:** Appendix numbering \& constants
>
> Thanks for asking this! For the clarity of presentation, the results (Theorems 2, 3, and 4) in the main body of the paper hide logarithmic terms appearing in the terms $C_{\xi}, C_{\bar{z}}, C_{\hat{x}}$, and $\Lambda$. Second, the logarithmic term appearing in the lower bound on $L$ in Theorem 4 hides logarithmic terms appearing inside logarithm. Lastly, we use a cleaner upper bound on the terms appearing inside the logarithm in $\Lambda$, and $L$. If the reader is interested in the details, we have presented the complete version of Theorems 2, 3, and 4 in the Appendix.
>
> Please note that, it is very common in the system identification and statistical learning theory literature to hide the logarithmic terms for the clarity of presentation [2,14,15].
>
> We have added a remark on this after Theorems 2, 3, and 4, and have re-numbered Theorems 5, 8, and 9 in the Appendix to Theorem 2 (complete version), Theorem 3 (complete version), and Theorem 4 (complete version) respectively.
>
> > **Q3:** Known latent dimension
>
> We respectfully disagree with the reviewer. Our main results (Theorems 1 – 4) do not require the true dimension $n$ of the latent space to be known. The error bounds in Theorems 1 – 3 depends on the complexity of the function class $\mathcal{F}$, which is captured by the cardinality of the $\epsilon$-covering set $\mathcal N$ (see Lemma 4 in the Appendix).
>
> Although, the robustness condition in Theorem 4, is written in terms of $\sigma_n (\mathcal{O} \mathcal{O}^\top)$, and $\sigma_n (\mathcal{C} \mathcal{C}^\top)$, it is worth noting that Theorem 4 assumes that $\mathcal{O}$ has full column rank, and $\mathcal{C}$ has full row rank. Hence, the robustness condition in Theorem 4 can be interpreted as, requiring the estimation error (Theorem 3) to be smaller than the smallest eigenvalues of $\mathcal{O} \mathcal{O}^\top$, and $\mathcal{C} \mathcal{C}^\top$. This condition can be easily satisfied by choosing the trajectory length $T$ to be large enough as we have shown below in our response to **Q5**.
>
> > **Q4:** Unify with the eigenvalues
>
> Thanks for the suggestion!  We have replaced $\sigma_n(\cdot)$ with $\lambda_{\min}(\cdot)$ in Theorem 4, and its proof in Appendix E.
>
> > **Q5:** Robustness condition
>
> Great question! We can combine Theorems 3, and 4, to get the robustness condition in terms of a requirement on the trajectory length $T$. Let $\bar \lambda_\min = \min \\{ \lambda_\min(\mathcal{O} \mathcal{O}^\top), \lambda_\min(\mathcal{C} \mathcal{C}^\top) \\}$.  Then, we can get rid of the the assumption that $2 ||\hat G_2 \hat G_1 - \mathcal{O} \mathcal{C} || \leq \bar \lambda_\min$ by choosing $ T \gtrsim \frac{T_0}{  \lambda_\min(\Sigma[\bar z_L]) \bar \lambda_\min^2 }$,  where $T_0$ is the numerator of the parameter estimation error upper bound (Theorem 3)
>
> > **Q6:** Same equation
>
> Thanks for catching this typo. We have fixed it!
>
> > **Q7:** Smaller training error
>
> We get the smaller training error at the true dimension because of two reasons: (1) implicit regulation because of weight decay, and (2) we have more number of samples than the parameters to be trained (unlike over-parameterized networks). In these settings, it has been previously observed that architecture search over the hidden state converges to the true dimension [12,13].
>
> > **Q8:** First experiment
>
> Thanks for asking this! To avoid confusion, we repeated Experiment 1 with architecture search (without explicitly setting $h=n$), and got similar plots. This is what we expected because it is evident, from Table 1, that the architecture search converges to the true dimension $h=n$. We will add this experiment when revising our manuscript!.
>
> # References
>
> see response to **Reviewer tePB**

---

> > ### Author Rebuttal · Reviewer_sRoF · 2026-04-03
> >
> > I am satisfied with the authors’ responses, and the proposed revisions will improve the paper.

---

> > > ### Author Response · Authors · 2026-04-07
> > >
> > > We sincerely thank the reviewer for increasing the score, and acknowledging that our initial rebuttal has fully resolved the concerns raised by the reviewer!
> > >
> > > We sincerely appreciate the reviewer's questions and/or suggestions regarding the related work, theorem/appendix numbering, hidden constants, latent dimension, unifying eigenvalues, smaller training loss, and robustness condition. We will incorporate these suggestions into the revision, and we believe these suggestions will improve the clarity and presentation of the paper.

---

### Official Review · Reviewer_WBH7 · 2026-03-24

**Soundness:** 3
**Presentation:** 3
**Significance:** 3
**Originality:** 3
**Overall Recommendation:** 4
**Confidence:** 2

**Summary:**

This paper investigates whether simple predictive models can successfully learn hidden-state representations relying solely on sequential observation data. Focusing on partially observed linear dynamical systems, where the algorithm only accesses noisy inputs and outputs rather than the evolving latent state, the authors bypass the traditional two-step approach of system identification followed by filter construction. Instead, they explore whether the filtering process can be learned directly through end-to-end empirical risk minimization. To test this, they train a two-layer linear auto-regressive model to predict future output windows based on a finite history of past inputs and outputs.

The primary contribution of this work is a theoretical proof demonstrating that the model's learned hidden representations align with Kalman filter state estimates (up to a similarity transform), despite lacking access to true system parameters or latent states. The authors establish this through three key pillars: proving that finite-history auto-regressive models can approximate Kalman filters, showing that the nonconvex optimization landscape is benign (featuring global minima and strict saddles), and establishing finite-sample guarantees for prediction, parameter estimation, and recovery errors. Synthetic experiments validate these claims, showing strong alignment with Kalman estimates, improved recovery with larger datasets, and optimal performance when the hidden dimension matches the true state dimension. Ultimately, this work proves that latent-state estimation organically arises from end-to-end predictive training.

**Compliance With Llm Reviewing Policy:**

Affirmed.

**Final Justification:**

The authors have adequately addressed my concerns and hence I will keep my score.

**Key Questions For Authors:**

1. What are technical challenges in going from a steady state Kalman filter to a time varying one?
1. Similarly, what are technical challenges in going from a two layer to a multi layer model?

**Limitations:**

The three limitations are the steady state Kalman filter assumption, the two layer model assumption as well as the lack of convergence results for the constrained problem are not given. However, it is to be noted that these limitations can be said to future works.

**Strengths And Weaknesses:**

**Strengths**

Is (to my knowledge) the only work that provides an end-to-end representation-learning result for partially observed LDSs: a trained two-layer AR model not only predicts well, but its hidden layer recovers Kalman-like latent states. That makes the result conceptually important for representation learning: it gives a rigorous example where next-step prediction really does recover meaningful hidden structure rather than just surface correlations. Theoretical analyses showing the convergence of the proposed methods are also given, which are novel as well.

The paper is well-organized as it lays out problem setup, two-layer architecture, Kalman filter target and the latent state recovery models given in a logical and easy to follow manner.

The results appear to be sound without any major noticeable errors.

**Weaknesses**

The result as of now seems to be only restricted to linear Gaussian partially observed systems and the steady-state Kalman-filter regime. In real life applications such assumptions might not be valid. Additionally, sticking to two layer models also reduces the real world applicabilty of the work.

---

> ### Author Rebuttal · Authors · 2026-03-31
>
> Thank you for your valuable review of our paper!
>
> > **Q1:**  Time varying Kalman filter
>
> Great question!
>
> First, under Assumption 2, the Kalman filter converges exponentially fast to the steady state [1]. Therefore, if the Kalman filter does not start in the steady state, it will converge after a small burn-in time $T_0$. Thus, without loss of generality, it is very common in statistical learning for control literature to assume that the Kalman filter starts in a steady state [2–5].
>
> Second, our results can be easily extended beyond the steady state Kalman filter. One important observation is that  when the process and measurement noises are i.i.d. Gaussian, then the innovation errors $e_t$ are independent Gaussian with time-varying covariance matrix $\Sigma_t$, which under the observability assumption (see Theorem 4) can be uniformly upper bounded by a matrix $\bar\Sigma \succ 0$ [7, Chapter 7] . Hence, we  can still solve the ERM problem (3.4) and show that our main results in Theorems 1 – 4  hold with an additional term appearing in our error bounds that decays exponentially with the burn-in time $T_0$. This can be understood from Eqn (4.1), which can alternately be written as,
>
> $$
> \hat x_t=\mathcal C_t \bar z_t+\bar A_{t-1} \bar A_{t-2} \cdots \bar A_{t-L} \hat x_{t-L}=\mathcal C \bar z_t+\bar A^L \hat x_{t-L}+(\mathcal C_t-\mathcal C ) \bar z_t+ (\bar A_{t-1}\bar A_{t-2} \cdots \bar A_{t-L} - \bar A^L) \hat x_{t-L},
> $$
>
> Where $\mathcal C_t$ is a time-varying extended controllability matrix constructed in a similar way as we constructed $\mathcal C$ in Section 4.1, albeit with time-varying matrices $\bar A_t$, and $F_t$.  Using exponential convergence of  Kalman filter to its steady state, we can show that, the last two terms in the above equation decays as $\alpha^{t-L}$ and $\rho^L$ respectively for some $\alpha, \rho \in (\rho(\bar A), 1)$. Combining these observations with our current analysis, we can make our results hold beyond steady state Kalman filtering, with  an additional term due to non-steady state initialization. However, this term decays exponentially with the burn-in time $T_0$.
>
>
> > **Q2:**  Multi layer model
>
> Great question! Extending our analysis to a deeper multi-layer linear architecture is indeed challenging. In the two-layer case, the hidden representation naturally appears at the single hidden layer activations $G_1\bar z_t$. In a multi-layer model $G_m \cdots G_1\bar z_t$, however, there are many intermediate representations, and it is not  immediately clear which layer, if any, should correspond to the Kalman state.
>
> Technically, in deeper linear models, even if one proves that the product $G_m\cdots G_1$ recovers the correct Hankel matrix $\mathcal{O}\mathcal{C}$, this does not by itself imply that any intermediate partial product $G_k\cdots G_1 \bar z_t$ recovers a meaningful latent state. This is a key difference from the two-layer setting, where the hidden layer is essentially the Kalman filtering representation, and can therefore be analyzed directly.
>
> A second challenge is optimization. In deeper linear networks, the loss landscape is no longer benign [9] (i.e., there exists bad saddle points). Thus, the optimization arguments used in our two-layer analysis do not transfer directly.
>
> For these reasons, extending our main end-to-end results (Theorem 4) from two layers to multiple layers would require new ideas beyond our current proof techniques. In particular, one would likely need additional structural assumptions to make the internal representation of a deeper network aligned with the Kalman filter representation. We verify this through numerical experiments with a 4-layer network as follows:
>
> For the random system described in the paper, with $n=4$, we define a 4-layer model with hidden dimensions  4, 8, and 12. For each hidden layer, we apply linear regression to find the correct coordinate of the system. Instead of scatter plot, we provide a $R^2$ statistic to show the linear relation between the regressed learned activations and the Kalman filter state.
>
> ### Layer 1 (dim=4)
> | |s1|s2|s3|s4|mean $R^2$|
> |-----|-----|-----|-----|-----|-----|
> |$H=1$|0.997|0.996|0.999|0.998|0.998|
> |$H=5$|0.989|0.993|0.997|0.996|0.994|
> ### Layer 2 (dim=8)
> | |s1|s2|s3|s4|mean $R^2$|
> |-----|-----|-----|-----|-----|-----|
> |$H=1$|0.997|0.996|0.999|0.998|0.998|
> |$H=5$|0.989|0.993|0.997|0.996|0.994|
> ### Layer 3 (dim=12)
> | |s1|s2|s3|s4|mean $R^2$|
> |-----|-----|-----|-----|-----|-----|
> |$H=1$|0.997|0.996|0.999|0.998|0.998|
> |$H=5$|0.989|0.993|0.997|0.996|0.994|
>
> We observe that the $R^2$ statistic barely changes over the layers. As long as the rank of each hidden layer is at least $n$, then the linear regression gives us almost the same predicted state estimate.
>
> > **Q3:**  Gaussian noise
>
> Please see our response to **Q5 of Reviewer vqtk**
>
> > **Q4:**  Convergence
>
> Please see our response to **Q1 of Reviewer tePB**
>
> # References
>
> see response to **Reviewer tePB**

---

> > ### Author Rebuttal · Reviewer_WBH7 · 2026-04-04
> >
> > Thank you for your response to my question. My concerns have been addressed and I will keep my score.

---

> > > ### Author Response · Authors · 2026-04-07
> > >
> > > We sincerely thank the reviewer for acknowledging that our initial rebuttal has fully resolved the concerns raised by the reviewer!
> > >
> > > We appreciate the questions asked by the reviewer regarding the technical challenges in going from: (1) steady-state to time-varying Kalman filter, (2) two layer to multi-layer AR model, and (3) Gaussian to non-Gaussian noise.
> > >
> > > Regarding the time-varying Kalman filter or extension to the general Kalman filter, we refer the reviewer to **our latest response to Q7 of Reviewer xuxP** (Reply Rebuttal Comment by Authors).
> > >
> > > We will incorporate our response into the revision, and we believe it will significantly improve the clarity of the paper!

---

### Decision · Program_Chairs · 2026-04-30

**Decision:**

Accept (regular)

**Comment:**

While reviewers agreed that the paper is generally well-written and empirical results back up the theoretical formulation, they also raised concerns about the paper's significance, mostly related to the restrictive settings of the analysis (i.e. linear, steady-state regimens) and the limited audience for such a paper in a large conference such as ICML.